# Evolutionary Algorithm-Based Iterated Local Search Hyper-Heuristic for Combinatorial Optimization Problems

**Stephen A. Adubi [1], Olufunke O. Oladipupo [1,2] and Oludayo O. Olugbara [2,\***

[1] Department of Computer and Information Sciences, Covenant University Ota, Ota 112104, Nigeria
[2] MICT SETA 4IR Center of Excellence, Durban University of Technology, Durban 4001, South Africa
\* Correspondence: oludayoo@dut.ac.za

**Abstract:** Hyper-heuristics are widely used for solving numerous complex computational search problems because of their intrinsic capability to generalize across problem domains. The fair-share iterated local search is one of the most successful hyper-heuristics for cross-domain search with outstanding performances on six problem domains. However, it has recorded low performances on three supplementary problems, namely knapsack, quadratic assignment, and maximum-cut problems, which undermines its credibility across problem domains. The purpose of this study was to design an evolutionary algorithm-based iterated local search (EA-ILS) hyper-heuristic that applies a novel mutation operator to control the selection of perturbative low-level heuristics in searching for optimal sequences for performance improvement. The algorithm was compared to existing ones in the hyper-heuristics flexible (HyFlex) framework to demonstrate its performance across the problem domains of knapsack, quadratic assignment, and maximum cut. The comparative results have shown that the EA-ILS hyper-heuristic can obtain the best median objective function values on 22 out of 30 instances in the HyFlex framework. Moreover, it has achieved superiority in its generalization capability when compared to the reported top-performing hyper-heuristic algorithms.

**Keywords:** combinatorial optimization; evolutionary algorithm; heuristic algorithm; knapsack problem; local search; maximum-cut problem; quadratic assignment

## 1. Introduction

Hyper-heuristics are search methodologies for solving numerous forms of combinatorial optimization problems (COPs) in routing applications [1,2], scheduling [3,4], machine learning [5,6], generation of solvers [7], and software engineering [8–10]. They have been applied to several other application domains of combinatorial optimization such as university examination timetabling, university course timetabling, and school timetabling problems [11]. Due to the countless peculiarities and inherent complexity of the different forms of optimization problems, a customized algorithm usually fails to generalize well. Hence, the impetus for studying hyper-heuristics over heuristics and metaheuristics is to address the problem of generality that hyper-heuristics provide across different forms of optimization problems [12]. Meta-learning for offline learning of heuristic sequences was recently described for solving capacitated vehicle routing and graph coloring problems [13]. The study [14] proposed a novel mechanism to create an effective recombination procedure for sub-trees in a genetic programming hyper-heuristic to solve the job-shop scheduling problem. A fuzzy logic-based method for the selection of low-level heuristics was described to solve many instances of the 0/1 knapsack problem [15]. A genetic programming-based hyper-heuristic was described for solving multiple tasks within a dynamic environment [16]. The ability of a generative hyper-heuristic underpinned by an artificial neural network was introduced for creating customized metaheuristics in

continuous domains [17]. The study in [18] proposed the use of a double deep Q-network (DDQN) for generating a constructive hyper-heuristic for COPs with uncertainties.

Hyper-heuristics can broadly be classified into selection and generation categories. Selection is employed to automatically control the use of low-level heuristics (LLHs) while generation is used to generate new LHHs from the building blocks of previous ones [12]. Manifold hyper-heuristics have been proposed in the literature, but a particular set of them have been developed and their performances benchmarked against others within the hyper-heuristics flexible framework (HyFlex) [19]. The HyFlex framework was initially used for the CHeSC 2011 competition where twenty algorithms were tested and their performances compared on six problem domains. The AdapHH hyper-heuristic [20] emerged as the winner of the competition after obtaining the highest rank in three domains of Boolean satisfiability (SAT), bin packing (BP), and the traveling salesman problem (TSP). The framework eventually became useful for benchmarking the performance of a newly proposed hyper-heuristic. The author of a newly proposed hyper-heuristic algorithm would conventionally compare its performance with the original 20 CHeSC entries. The fair-share iterated local search (FS-ILS) hyper-heuristic utilized a speed-proportional selection scheme (SpeedNew) as the heuristic selection mechanism and accept probabilistic worse (APW) as the solution acceptance mechanism [21]. It selects from a pool of perturbative heuristics during the perturbation phase of the ILS. It applies the local search heuristics in a variable neighborhood descent (VND) fashion on the resultant solution from the perturbation phase until the solution can no longer be improved. The VND is a variant of the variable neighborhood search (VNS) metaheuristic that deterministically explores the neighborhoods in a search space.

The FS-ILS outperformed the 20 CHeSC entries in the HyFlex framework by obtaining the highest scores on the Boolean satisfiability (SAT), permutation flow-shop (PFS), and vehicle routing problem (VRP) domains. A Thompson sampling hyper-heuristic (TSHH) was proposed in [22] and uses the Thompson sampling learning algorithm to respectively select perturbative and local search heuristics during the perturbation and intensification phases of the iterated local search (ILS). The selection of a heuristic to apply was based on its record of successes and failures when applied to a given problem. If the combined effect of a perturbative heuristic and a local search heuristic (when applied in an iteration) leads to a new best global solution, the "success" count of the heuristics is incremented; otherwise, the "failure" count is incremented, which makes it a reinforcement learning scheme. The TSHH implemented an improving-or-equal (IE) acceptance mechanism and finished second behind the AdapHH using the F1 ranking. The hyper-heuristic outperformed in personnel scheduling (PS), PFS, and TSP and underperformed in SAT and VRP. Ferreira et al. [23] proposed different settings of a multi-armed bandit scheme for heuristic selection, with the best version being able to only do well on VRP problems, which the authors blamed on the hyper-parameter settings.

The HyFlex framework was extended in [24] by introducing three new problem domains of the knapsack problem (KP), quadratic assignment problem (QAP), and maximum-cut problem (MAC). The experimental procedure in the work pitted FS-ILS against a no-restart version of FS-ILS (NR-FS-ILS), AdapHH, evolutionary programming hyper-heuristic (EPH), a competitor in CHeSC 2011, and two other simple random procedures that differ in their acceptance mechanisms. EPH is based on the principle of evolutionary programming and co-evolution that concomitantly maintains a population of solutions with a population of low-level heuristic sequences that are applied to the solutions [11]. The AdapHH emerged as the winner on these new domains and the random procedure that accepts all moves surprisingly finished second while FS-ILS finished fifth among the six hyper-heuristics compared. In addition, the SSHH was tested on these three new problem domains [25] and its performance was compared with those of others presented for the extended HyFlex framework [24]. The SSHH and AdapHH emerged as the best two hyper-heuristics based on the experimental results. The new problem domains were used in two other works [26,27] that tuned a memetic algorithm, but only [27] compared the

proposed hyper-heuristic with others that were tested on the domains, using 15 out of the entire set of 30 benchmarking instances.

This study was inspired by the inherent weaknesses exhibited by ILS hyper-heuristics on the extended HyFlex suite to design an evolutionary algorithm-based iterated local search (EA-ILS) hyper-heuristic. The following two critical points are highlighted as areas of concern for ILS hyper-heuristics. First, ILS hyper-heuristics utilizing a typical local search invocation based on VNS may not be effective when solving problems that require deep search space and are time-consuming [28]. Therefore, designing an improved local search is worth exploring [29]. Second, although hyper-heuristic methods are generally not exact, an alternative for covering a deeper search space for performance enhancement could be to design ILS-based hyper-heuristics to perturb solutions more than once before the local search invocations. This will depend on how effective these multiple perturbations can be for solving a COP. Since the perturbation strength is prime to the performance of ILS [30,31], designing a hyper-heuristic with an option to perturb solutions more than once is very viable.

The introduced hyper-heuristic algorithm works on the principles of the basic ILS and evolutionary algorithms. It uses a novel mutation operator to construct LLH sequences with the possibility of multiple perturbations that ultimately end with a local LLHs search. The ILS is widely used in the literature for solving a wide range of COPs such as orienteering [32], inventory routing [33], classical knapsack [34], vehicle routing [35–37], and course timetabling [38–40]. The metaheuristic appears to be a versatile method that has been frequently combined with other optimization algorithms to solve COPs. Different instances of this hybridized set-up include pairing ILS with quadratic programming [41], tabu search [42], evolutionary algorithms [35], simulated annealing [43], and tabu search with simulated annealing [44]. The EA-ILS hyper-heuristic was fortified with a local search module based on the connotation of the hidden Markov model (HMM) to automatically learn promising sequences of local search heuristics rather than exhaustive application. The following are the unique contributions made in the present study to the discipline of combinatorial optimization:

- The application of a novel mutation evolutionary operator to construct promising perturbative heuristic sequences of variable length of 1 or 2 to address a weakness of the previous ILS-based hyper-heuristics is an important contribution of the present study;
- The design of the EA-ILS hyper-heuristic algorithm that combines the capability of ILS with a specialized mutation evolutionary operator for improved performance in solving numerous COPs is a unique contribution of this study;
- The experimental comparison of the EA-ILS hyper-heuristic with the existing hyper-heuristics in the HyFlex framework to demonstrate the effectiveness of the introduced algorithm is a distinctive contribution.

The remainder of this paper is succinctly summarized as follows: Section 2 reviews the relevant literature by surveying the recent methods that have been applied to tackle the problems of knapsack, quadratic assignment, and maximum cut. Section 3 presents the materials and methods used in the present study and explicates the EA-ILS hyper-heuristic algorithm. Section 4 discusses the experimental results of evaluating the performance of the EA-ILS hyper-heuristic against the performances of the existing hyper-heuristics in the HyFlex framework. The article is ultimately concluded in Section 5 by summarizing the important highlights.

## 2. Related Studies

There are numerous algorithms reported in the literature for solving the COPs of knapsack, quadratic assignment, and maximum cut. Previous studies have agglutinated different algorithms within the ILS for performance improvement. The authors in [45] employed meta-learning to improve the performance of ILS on the Google machine

reassignment problem (GMRP). The idea was to learn from some instances of a problem and recommend suitable ILS components based on the instance being solved. Hu et al. [46] combined a genetic algorithm (GA) with ILS to solve the instances of a dominating tree problem (DTP). In their approach, multiple solutions were kept and each solution in the population of solutions was subjected to possible improvement through the ILS procedure. The GA presented in the study utilized a specialized mutation procedure with high diversification strength and ignored crossover operators. The specialized mutation operator was applied after the ILS phase to each solution in the population during the second stage of the search process. The second stage helps to improve diversity in the solutions returned by the ILS in the first stage according to the study.

The study reported in [47] combined the ILS with other techniques to effectively solve numerical optimization problems. The perturbation strategy was based on the success-history-based parameter adaptation for differential evolution (SHADE) while the local search phase was based on a mathematical model. The ILS keeps multiple solutions while it relies primarily on the perturbation mechanism of the SHADE to control the strength of the ILS perturbation. The partition crossover operator was incorporated into an ILS framework for the computational design of proteins [39]. The crossover operator was utilized as an additional perturbation operator to combine two solutions generated by the ILS framework to produce a new solution that is further enhanced by the steepest descent algorithm. The algorithm contested favorably with a classical ILS and the Rosetta fixbb method. The APW is not a new acceptance mechanism employed in this paper [21,48,49]. However, a strategy is proposed to oscillate its important parameter called temperature during the search process.

In [50], ant colony optimization (ACO) was employed to generate rules for the selection of heuristics for the knapsack problem. Candidate hyper-heuristics were constructed by the virtual ants over time using the current problem state until all items had been packed. The best hyper-heuristic constructed during the simulation can now be applied to unseen instances. The studies in [51,52] designed a feature-independent hyper-heuristic through an evolutionary algorithm to solve the 0/1 knapsack problem. Similar to the approach in the previous study [50], a set of training instances were used to construct viable hyper-heuristics that were superior to the individual LLHs of the problem investigated. Olivas et al. [15,53] incorporated fuzzy logic into the inference process of the selection of heuristics for the knapsack problem. The authors considered seven features of the problem as inputs to the fuzzy inference engine and four LLHs as outputs. The fuzzy-based hyper-heuristic was optimized by a genetic algorithm that was benchmarked against traditional hyper-heuristics optimized by a particle swarm optimization (PSO) algorithm. The other optimization methods not based on hyper-heuristics that have been applied to solve the knapsack problem include the binary monarch optimization algorithm [54], binary bat algorithm [55], whale optimization [56], list-based simulated annealing [57], and artificial bee colony optimization [58]. A recent study [59] compared the genetic algorithm (GA), simulated annealing (SA), dynamic programming (DP), branch-and-bound (BB), greedy search (GS), and a hybrid of GA–SA to solve the knapsack problem.

Senzaki et al. [60] applied a hyper-heuristic method based on the choice function selection mechanism to solve the multi-objective quadratic assignment problem (mQAP). The task of the choice function is to select the genetic operators that are LLHs during an iteration of the multi-objective evolutionary algorithm (MOEA) as the base method. The method was reported to perform quite well on 22 instances of mQAP after being benchmarked against other multi-objective optimization algorithms. Some of the approaches for solving a quadratic assignment problem (QAP) tend to favor the hybridizations of two or more algorithms. The study [61] hybridized quantum computing principles with an evolutionary algorithm to solve QAP while another study [62] integrated tabu search with a whale optimization algorithm. In [62], tabu search was employed to improve the solution constructed by the whale optimization algorithm. The study [63] hybridized an evolutionary algorithm called elite GA with a tabu search for solving 135 instances of the QAP

from the well-known QAPLIB dataset. The authors reported that the hybrid method obtained the best-known solutions for 131 instances. Dokeroglu et al. [64] applied a robust tabu search method to control the exploration–exploitation balance within an artificial bee colony. The resulting method was able to optimally solve 125 out of the 134 benchmarking instances of the problem. The agglutination of GA with Tabu (GA–Tabu) search has been recently published for solving the QAP [65] with a favorable comparative performance.

Tabu search was hybridized with the evolutionary algorithm to solve the MAC [66]. The authors were able to find the best new solutions for 15 of 91 instances used to test their algorithm. Chen et al. [67] proposed a binary artificial bee colony algorithm with a local search procedure to solve 24 instances of the MAC as reported in the literature. Kim et al. [68] compared the performances of harmony search and two implementations of genetic algorithms, namely generational GA and steady-state GA, that were tested on 31 instances of the MAC. The study reported that the harmony search algorithm outperformed the two GA implementations. The Q-learning model-free reinforcement learning algorithm was proposed to solve some instances of the MAC [69]. The method utilizes a message-passing neural network (MPNN) to predict the Q-values of removing or adding vertices to solution subsets. Seven observations were configured to characterize the problem state during the training of the deep Q-network.

Previous algorithms have used the same instances of the HyFlex framework for performance evaluation as follows: The FS-ILS with its variant NR-FS-ILS, based on the ILS, recorded a subdued performance on the extended HyFlex domains [24]. The AdapHH [20] is a state-of-the-art hyper-heuristic algorithm that outperformed both FS-ILS and NR-FS-ILS on the extended benchmarking test domains [24,25]. It uses a relay hybridization technique to pair LLHs while solving a problem instance and emerged first in the CHeSC 2011 competition. The sequence-based selection hyper-heuristic (SSHH) based on the principles of the HMM was proposed in [70]. It attempts to automatically identify the optimal sequences of heuristics while optimizing the solution of a given instance of a problem domain. The SSHH was initially tested on the first six problem domains in the first version of the HyFlex framework to outperform the twenty CHeSC 2011 entries in the SAT, BP, and TSP problem domains. The algorithm recorded a favorable performance amongst several competitors on KP, QAP, and MAC [25]. The other methods include the EPH and two simple hyper-heuristics that choose LLHs randomly [24].

It is important to review the past research works on the HyFlex domains to identify inconsistencies for further improvement. However, algorithms such as the SSHH, and sometimes AdapHH, that were designed to automatically generate heuristic sequences could benefit from the explicit separation of diversification and intensification found in ILS-based methods. The ILS-based methods applied on the extended HyFlex domains have not yet been generally effective. It is still valuable to devise new means of exploring the strengths of the ILS methods because of the limited evidence of its effectiveness as a hyper-heuristic [21,22,48,71,72]. The present study has employed the instances provided in the extended HyFlex framework [24] to test the generality of the EA-ILS hyper-heuristic.

## 3. Materials and Methods

The materials used to conduct this study and descriptions of the main COPs investigated are presented in this section. Descriptions of the methods applied to construct the EA-ILS hyper-heuristic that solves the investigated COPs are provided thereafter.

### 3.1. Materials

The instances used for testing the EA-ILS hyper-heuristic contain extensions to the original HyFlex dataset located on the webpage (https://github.com/Steven-Adriaensen/hyflext accessed on 10 September 2022). They are executable files of the implementations of the instances of new problems added to the HyFlex framework. The webpage of all the instances has the raw data for the existing hyper-heuristics used to benchmark the

performance of the EA-ILS hyper-heuristic. There are a total of 30 instances available in the extended HyFlex test suite for benchmarking the performance of a hyper-heuristic. All algorithms were coded in the Java programming language. The computer used for testing the EA-ILS hyper-heuristic has 8 gigabytes (GBs) of random access memory (RAM) and an Intel i5-3340 M central processing unit (CPU) with a 2.70 GHz clock speed. The evaluation metrics employed for benchmarking the performance of the EA-ILS hyper-heuristic with the existing comparative algorithms are the standard evaluation metrics of μ-norm, μ-rank, best, worst [24,25], F1 scoring system [23,49,73], and statistical evaluation of the Friedman test [74] and boxplot analysis [22,75]. The ten benchmarking instances of the problem available in the test suite [24] were all taken from the well-known QAPLIB dataset [76]. The mathematical formulation used in the current study for the MAC problem is based on a recent reformulation [77]. The number of low-level heuristics per category for each problem is presented in Table 1. Crossover heuristics are ignored in the table because they were not used by the EA-ILS hyper-heuristic.

**Table 1.** Tally of the LLHs across the Knapsack, Maximum-Cut, and Quadratic Assignment Problems.

| Problem | Mutation | Ruin-Recreate | Local Search | Total |
|---|---|---|---|---|
| Knapsack problem (KP) | 5 | 2 | 6 | 13 |
| Quadratic assignment problem (QAP) | 2 | 3 | 2 | 7 |
| Maximum-cut problem (MAC) | 2 | 3 | 3 | 8 |

*3.2. Methods*

A brief discussion of the basic ILS and a detailed description of the EA-ILS hyper-heuristic are provided in this section to illuminate the novelty of the proposal. The differences between the two hyper-heuristics are alluded to show the contribution made by the ILS.

3.2.1. Basic Iterated Local Search Algorithm

The iterated local search (ILS) [78] is a simple approach based on the principles of diversification (perturbation) and intensification (local search). The search approach outlined in Algorithm 1 switches between the diversification and intensification phases throughout the search process. The initial solution of the ILS can be generated by any constructive heuristic that is suitable for the given problem domain. In the iterative block, perturbation of the incumbent solution is performed to diversify the search and avoid circling a particular search area. If the perturbation operation is too strong, the search process cannot be controlled, and it becomes a random restart algorithm. Conversely, a shallow perturbation can perform searches to keep revisiting a particular search area, thereby limiting the progress of the search. In a typical circumstance, the perturbation phase of the search worsens the incumbent solution, but the local search phase is designed to search for better solutions within the neighborhood of the perturbed solution. The solution returned from the local search is considered a candidate solution when it is accepted by the acceptance mechanism. If the candidate solution is accepted, it replaces the incumbent solution, and the search continues in that fashion until a stopping criterion is met.

---

**Algorithm 1:** Basic Iterated Local Search

1.   $S_0 \leftarrow$ generateInitialSolution ()
2.   $S \leftarrow$ perform a local search on $S_0$
3.   **while** ¬stopping_condition
4.      $S' \leftarrow perturb(S)$
5.      $S'' \leftarrow$ perform a local search on $S'$
6.      $S \leftarrow \begin{cases} S'', & if\ S''\ is\ accepted \\ S, & otherwise \end{cases}$
7.   **end while**

---

### 3.2.2. Evolutionary Algorithm-Based Iterated Local Search

The evolutionary algorithm-based iterated local search (EA-ILS) hyper-heuristic searches the space of LLHs during the search process. It does so by constructing and editing the sequences of LLHs to conform to the working principles of the ILS. Perturbative LLHs are applied, followed by the application of the local search LLHs to affect intensification in ILS. Similarly, the sequences generated by the EA-ILS hyper-heuristic begin with perturbative LLHs and end with a local search terminal. The conception of "operation sequence or simply a sequence" is used because the sequences are typically longer than what is presented when applied to a solution. Figure 1 depicts a sample sequence where EA-ILS applies LLH5 to the incumbent solution. The resulting solution is further perturbed with another perturbative heuristic (LLH7), and the local search heuristics are applied for intensification thereafter. The last operation, which is a call to the local search (LS) module, could use more than one local search LLH. Hence, the sequence of LLHs applied in an iteration could be longer than 3. The maximum length of a sequence is set at 2, and it is the third member of the sequence that represents the local search terminal. This is to keep the length of time dedicated to applying the sequence of LLHs on a solution during an iteration at a reasonable amount. The operations that appear in the EA-ILS hyper-heuristic described by Algorithm 2 are explained in Table 2. The improving iteration is an iteration of the EA-ILS hyper-heuristic that leads to the update of the best global solution. The non-improving iteration is an iteration of the EA-ILS hyper-heuristic that does not lead to the update of the best global solution. The acceptance strategy is the value of the temperature parameter used in the APW in the acceptance mechanism of this study.

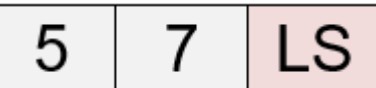

**Figure 1.** Sample operation sequence in the EA-ILS hyper-heuristic.

**Table 2.** Description of Operations of the EA-ILS hyper-heuristic.

| Operation | Description |
|---|---|
| X.add(x, L) | Add a new member $x$ to a bounded list X with bound size L such that when a new entry updates the size of X to L + 1, the item at the top of the list is removed. |
| X.add(x, L, t) | The addition of $x$ from the previous operation is repeated $t$ times. |
| H | An operation sequence or simply a sequence. |
| A | The bounded list for storing the most recent sequences that improved the best global solution. |
| Rnd.Real(x, y) | This operation generates a random real number $\delta$ such that $x < \delta < y$. |
| Rnd.Int(x, y) | This operation generates an integer $\delta$ such that $x \leq \delta < y$. |
| Rnd.Bool() | Randomly generate a Boolean variable. |
| better(x, y) | Denotes a comparison operation that returns a Boolean value depending on whether solution $x$ is better than $y$ based on their objective function values. |
| $P_w^1$, $P_w^2$ | Bounded lists for recently rewarded parameter values for the LLHs. The former is kept for perturbative LLHs while the latter is kept for local search LLHs. |
| $P_f^1$, $P_f^2$ | The fixed lists of all possible parameter values available for the perturbative and local search LLHs, respectively. |
| $L_p$ | The bound size for the $P_w^1$ and $P_w^2$ lists. |
| $A_t$ | The number of consecutive non-improving iterations allowed for the acceptance strategy. Experimental value = 15. |
| $H_t$ | Similar to $A_t$, but applied to the operation sequences. Experimental value = 1. |
| nA | The number of non-improving iterations completed so far for an acceptance strategy. |
| nH | Similar to nA, but applied to the operation sequences. |
| $imp_A$ | The number of improving iterations completed so far for an acceptance strategy. |
| $imp_H$ | Similar to $imp_A$, but for LLHs |

| | |
|---|---|
| $L_{add}$ | A pre-determined list of possible values that could be added to the value of the current acceptance strategy during its mutation. |
| $P_A$ | A bounded list of rewarding acceptance strategies based on the improvement of the global best solution. |
| $H_{pert}$ | A list of combinations of mutation and ruin-recreate LLHs perturbative heuristics for a problem domain. |
| rndSel() | A function that returns a random member of a list. |
| A | The current value of the temperature parameter being used by the acceptance mechanism. |
| cur_a | The currently engaged acceptance strategy. |
| $S_b$ | The best solution found so far. |
| $S''$ | The proposed solution after a perturbation–intensification cycle. |
| $S_0$ | The incumbent solution during the search process. |

---

**Algorithm 2:** The EA-ILS hyper-heuristic

Variable: runtime

1. $S_0 \leftarrow$ generateInitialSolution()
2. $S_b \leftarrow S_0$
3. init()
4. **while** getElapsedTime() < runTime
5.    $H \leftarrow$ rndSel($A$)
6.    **for** i $\leftarrow$ 1 to p + 1   ▷ $p$ is the number of offspring
7.      **if** i > 1
8.        H $\leftarrow$ mutate(H)
9.      **end if**
10.     setParam()
11.     **while** nH < $H_t$
12.       $S'' \leftarrow$ Apply the operation sequence $H$ on $S_0$
13.       set_accept_strategy()
14.       $S_0 \leftarrow \begin{cases} S'', & \text{if } S'' \text{ is accepted, goto line 15} \\ S_0, & \text{otherwise, goto line 24} \end{cases}$
15.       updateParam() ▷ if $S''$ replaces $S_0$
16.       **if** better($S_0$, $S_b$)
17.         $S_b \leftarrow S_0$
18.        updateLS()
19.        $imp_H \overset{+}{\leftarrow} 1$
20.        $imp_A \overset{+}{\leftarrow} 1$
21.        $nA \leftarrow 0$
22.        $nH \leftarrow 0$
23.       **else**
24.        $nA \overset{+}{\leftarrow} 1$
25.        $nH \overset{+}{\leftarrow} 1$
26.       **end if**
27.     **end while**
28.     **if** $imp_H > 0$
29.       A.add(H, $|H_{pert}|$, $imp_H$) ▷ add $H$ to archive $A$
30.     **end if**
31.    **end for**
32. **end while**

The initial solution for the EA-ILS hyper-heuristic is generated in Line 1 of Algorithm 2 through a constructive LLH of a given problem domain. In Line 2, the initial solution is used to initialize the best solution found so far. The initial population of sequences is generated by the init() procedure in Line 3 and stored in a bounded list ($A$) of size $|H_{pert}|$; each one is similar to the sequence shown in Figure 1. In the same line, some other tasks to initialize some values needed within the algorithm are carried out. They include the following:

- Initialization of the bounded lists for parameters $P_w^1$ and $P_w^2$ such that each element in the initial lists $P_f^1$ and $P_f^2$ is represented at least once within $P_w^1$ and $P_w^2$, respectively. The size of these bounded lists at the initialization stage is $L_p = 5$;
- Initialization of the bounded list for acceptance strategies ($P_A$) with a shuffled version of the array {0.38, 0.25, 0.15}. This means that the highest value of the temperature parameter for the accept probabilistic worse (APW) acceptance mechanism is initially set to 0.38;
- Randomly selecting a member of $P_A$ as the starting parameter value of the acceptance strategy before the search process begins.

The runTime variable is the time that is checked against the elapsed time during the run of the EA-ILS hyper-heuristic. The value set that is returned by a benchmarking program for runTime is reported in Section 4. The getElapsedTime() function returns the total elapsed time because the beginning of the EA-ILS hyper-heuristic runs on an instance of a problem. In Line 5, a sequence $H$ is randomly picked from the bounded list of sequences ($A$) initialized in Line 3. This bounded list is set up to keep the sequences that generated the best new solutions when they were applied. Line 6 is the loop that controls the mechanism of applying sequences and their mutations. There are only two iterations completed by the loop because the value of $p$ is set to 1 so that every selected sequence spawns an offspring that is tried before the selection of another sequence from archive $A$. The mutation evolutionary operator designed for the EA-ILS hyper-heuristic is applied to $H$ in Line 8 during the second iteration of the inner loop in Line 6 to likely produce another sequence.

The parameters for the perturbative LLHs in the sequence are set in Line 10. This only happens to the perturbative LLHs that use the *intensity of mutation* parameter. Line 11 marks a cycle of repeated application of an operation sequence until it no longer improves the global best solution $S_b$. In Line 12, the current operation sequence is applied to a solution by successively applying the perturbative LLHs in the sequence before the execution of the local search module for intensification. The acceptance strategy may be changed after applying a sequence. This will depend on whether the number of consecutive non-improving iterations allowed for the current acceptance strategy has elapsed or not, as in Line 13 of Algorithm 2. The value of the parameter to be applied for a perturbative or local search LLH is selected based on a random value through the function setParam() described by Algorithm 3. This random value is checked against the value of $\varphi = 0.5$. Consequently, the parameter is selected at 50% probability from a recency list of parameters with accepted solutions; otherwise, it is selected from the general list of a parameter value $P_f$. There are two versions of $P_f$: one is maintained for the perturbative LLHs while the other is maintained for the local search LLHs. The value of $P_f^1 = \{0.0, 0.1, \ldots, 0.6\}$ and that of $P_f^2 = \{0.5, 0.6, 0.7\}$.

| **Algorithm 3:** setParam() |
| --- |
| $\delta \leftarrow Rnd.Real(0, 1)$ |
| **if** $\delta < \varphi$ |
|   p $\leftarrow$ rndSel($P_w$) $\triangleright$ could be $P_w^1$ or $P_w^2$ |
| **Else** |
|   p $\leftarrow$ rndSel($P_f$) $\triangleright$ could be $P_f^1$ or $P_f^2$ |
| **end if** |
| applyParam(p) $\triangleright$ Apply the selected parameter value |

The incumbent solution $S_0$, as in Line 14 of Algorithm 2, is replaced by the proposed $S''$ if the acceptance mechanism accepts the proposed solution. In this scenario, Line 15 of Algorithm 2—which updates the parameter-bounded lists for the parameterized perturbative and local search LLHs—is triggered. The following activities, in the order they appear, are triggered only if a best new solution is found in the current iteration, and Line 16 of Algorithm 2 checks for this condition. The best solution found so far is replaced by the proposed solution and learning parameters for the local search procedure are updated. The number of consecutive *improving* iterations for both the current sequence and the acceptance strategy is incremented while the counters for their *non-improving* iterations are reset to zero, as in Lines 17–22 of Algorithm 2. In particular, Line 18 updates the data structures of the local search when a new best global solution is found by the hyper-heuristic. The full algorithmic outline of the local search procedure is given in Section 3.2.4. The numbers of non-improving iterations for both the currently engaged acceptance strategy and the sequence are updated in Lines 24 and 25 when the proposed solution is not accepted or is inferior to the global best solution, as indicated in Line 23 of Algorithm 2. The perturbed solution goes through the intensification phase when the execution of a sequence reaches the local search terminal as identified with the "LS" tag in Figure 1. The intensification phase is carried out through the LS-Seq local search procedure, and the resulting solution gets screened by the acceptance mechanism in Line 14 of Algorithm 2.

A different strategy is taken in the local search module of the EA-ILS hyper-heuristic for applying local search heuristics during the intensification phase. Instead of applying the heuristics in a variable neighborhood style as in [21,49], HMM is employed to automatically learn promising sequences of local search LLHs. A vector iScore of size n and an n × n matrix called pScore are maintained for the n local search LLHs. A roulette wheel procedure is employed to select the first local search LLH in an iteration of the local search phase based on the values of the iScore vector. Subsequent selections through a roulette wheel procedure are performed with the pScore matrix that captures the transition probability of selecting a given $j$ local search LLH after applying a previous one $i$.

Line 28 of Algorithm 2 updates the bounded list of high-quality sequences discovered during the search process. This update is achieved by adding the recently applied sequence to the bounded list if it has updated the global best solution at least once during its usage. In Line 29 of Algorithm 2, $imp_H$ copies of a sequence are added to the bounded list. This will happen if, at the end of the update operation, the number of operation sequences after the operation remains $|H_{pert}|$, regardless of the number of items added. The major features of the EA-ILS hyper-heuristic are explicated as follows:

- The sequence of perturbative LLHs is discovered by an evolutionary algorithm, which means that the EA-ILS hyper-heuristic does not use a mainstream selection mechanism'
- The *temperature* parameter of the acceptance mechanism, that is, the APW of the present study, oscillated during the search process;
- The intensification phase in most ILS-based hyper-heuristics such as the FS-ILS algorithm was achieved through a local search procedure based on the VND. The EA-ILS hyper-heuristic carries out intensification through a procedure based on the HMM.

The algorithmic details of LS-Seq and the local search procedure of the EA-ILS hyper-heuristic can be found in [48];

- The parameter values of the LLHs in the EA-ILS hyper-heuristic are learned over time using bounded lists.

The acceptance strategy implemented by Algorithm 4 for the search process represents the value of the temperature parameter used in the APW. The value is reviewed after $nA$ iterations have elapsed for the current acceptance strategy. A new acceptance strategy not in the bounded recency list of acceptance strategies denoted by the symbol $P_A$ is generated with a probability of $\theta = 0.3$ through the linear_mutation procedure invoked in the set_accept_strategy described by Algorithm 4; otherwise, a value is randomly selected from the recency list. If the current acceptance strategy improves the best global solution during the episode, it is added to the bounded list before it is replaced by the newly chosen acceptance strategy. The variables attached to an acceptance strategy are reset to zero. The size of the list $P_A$ is bounded by a value $L_A = 7$, which means that the size of the list will never exceed the value of 7 that was chosen after preliminary experiments.

---

**Algorithm 4:** set_accept_strategy()

**if** nA = $A_t$
  $\delta \leftarrow Rnd.Real(0,1)$
  **if** $\delta < \theta$
    a $\leftarrow$ linear_mutation()
  **else**
    a $\leftarrow$ rndSel($P_A$)
  **end if**
  **if** $imp_A > 0$
    $P_A$.add(cur_a, $L_A$)
  **end if**
  cur_a $\leftarrow$ a
  $imp_A \leftarrow 0$
  nA $\leftarrow$ 0
**end if**

---

The linear mutation operation that changes the value of an acceptance strategy during the execution of Algorithm 4 is detailed by Algorithm 5. Firstly, an acceptance strategy is randomly selected from the bounded list of "elite" acceptance strategies, a value p1_add is then randomly selected from a fixed list $L_{add}$ and added to the selected acceptance strategy. If necessary, a repair is made on the resultant value to make sure the output of the procedure is valid and is not greater than 1. In this study, $L_{add} = \{0.1, 0.2, \dots, 0.5\}$ was determined after a set of preliminary experiments.

---

**Algorithm 5:** linear_mutation()

p1 $\leftarrow$ rndSel($P_A$)
p1_add $\leftarrow$ rndSel($L_{add}$)
p2 $\leftarrow$ p1 + p1_add
**if** p2 > 1.0
  p2 $\leftarrow$ p2 − 1.0
**end if**
**Output:** p2

---

The procedure for updating the bounded lists of parameter values used by LLHs for both local search and perturbative categories is described by Algorithm 6. These lists are

updated once an ILS cycle produces a solution that is accepted by the acceptance mechanism. The while loop controls the update of the bounded list for perturbative LLHs, $P_w^1$. The variable $P_{iter}$ simply represents the number of iterations covered by the perturbation stage while $p_i$ represents the value used for the intensity of the mutation parameter when the *i*th perturbative heuristic in the operation sequence is applied to a solution. The value of $P_{iter}$ will be 1 if only one perturbative LLH appears in the operation sequence that was applied in the ILS cycle, or 2 if there are two perturbative LLHs in the operation sequence. The eventual number of updates on $P_w^1$ depends on whether the latest perturbative LLHs use the intensity of the mutation parameter or not. However, no parameter update is made for the perturbative LLHs if none of the perturbative LLHs applied in the last ILS cycle use the parameter. The local search parameter update begins after the while loop. The LS-Seq local search procedure keeps the parameter values of the local search LLHs applied during the intensification phase in a list tagged p_list. If the size of p_list is zero, it means that no single local search heuristic has improved the solution constructed during the perturbation stage. The for loop simply goes through the list and adds every member to the bounded list of "elite" parameters for the local search heuristics. The value of $L_p = 5$ means there is a maximum of five parameter values in either of the bounded lists at any point in time during the search process.

---

**Algorithm 6:** updateParam()

---

i ← 0
**while** i < $P_{iter}$
　$P_w^1$.add($p_i$, $L_p$)
　i ← i + 1
**end while**
**if** |p_list| > 0
　**for** $p \in p\_list$
　　$P_w^2$.add($p_i$, $L_p$)
　**end for**
**end if**

---

3.2.3. Evolutionary Operator of EA-ILS Hyper-Heuristic

The design of a novel mutation evolutionary operator for the EA-ILS hyper-heuristic is discussed in this section. The mutation operator that is applied to a sequence during the search process is implemented by Algorithm 7, which presents three main cases as follows.

1. **Wild mutation**: This is the first case; it occurs (pw ∗ 100) % of the time when mutation takes place. The value pw represents the probability of "wild mutation", which was set at 0.5 in this study. If {3, 2} is changed to {4, 0}, for example, it would be noticed that the two pairs are not similar, hence the name "wild mutation";

2. **Add random**: Since the maximum length of a sequence is 2, there are two cases when adding a new perturbative LLH to a sequence. **Case 2a**: If the randomly generated position (tagged *loc*) is 0, which denotes adding the new LLH at the first position, randomly select a perturbative LLH and add it to position 0, replacing the incumbent occupant of position 0. **Case 2b**: This is similar to Case 2a, only that the newly generated member is fixed at position 1;

3. **Remove random**: The remove random case selects a random position and the LLH at that position is removed or replaced. This case presents three possible sub-cases as follows: The first two sub-cases are triggered when the position to remove from is the first position, i.e., position 0. The last sub-case is when a LLH at position 1 is to be removed; in this sub-case, the LLH at this position is simply removed. **Case 3a**: If a sequence is full, i.e., there are two perturbative LLHs in the sequence, remove the LLH $h^r$ at position 0 and move the LLH at the next position to position 0. **Case 3b**:

The current perturbative LLH at position 0 in the sequence is replaced with randomly generated LLH while the second position is still vacant.

---

**Algorithm 7.** mutate()

---

**Input:** an operation sequence H

$\delta \leftarrow$ Rnd.Real(0,1)

**if** $\delta <$ pw                    ▷ **Begin Case 1**

  $H_0 \leftarrow$ rndSel($H_{pert}$)

  $H_1 \leftarrow$ rndSel($H_{pert}$)

  **Return**

**end if**                    ▷ **End Case 1**

loc $\leftarrow$ Rnd.Int [0, 2)

**if** Rnd.Bool()                    ▷ **Begin Case 2**

  h $\leftarrow$ rndSel($H_{pert}$)

  $H_{loc} \leftarrow$ h

**else**                    ▷ **Begin Case 3**

  **if** loc = 0 $\wedge$ $H_1 \neq -1$

    $H_0 \leftarrow H_1$

    $H_1 \leftarrow -1$

  **else if** loc = 0 $\wedge$ $H_1 = -1$

    h $\leftarrow$ rndSel($H_{pert}$)

    $H_{loc} \leftarrow$ h

  **else**

    $H_{loc} \leftarrow -1$

  **end if**

**end if**

---

3.2.4. Local Search Procedure of EA-ILS Hyper-Heuristic

The local search procedure (LS-Seq) for the intensification stage of the EA-ILS hyper-heuristic is briefly described in Algorithm 8. It is based on the hidden Markov model principle for constructing effective heuristic sequences [70] and was proposed as an alternative to the VNS [48]. The LS-Seq was designed to eliminate the excessive iterations in the ILS-based hyper-heuristics such as FS-ILS and NR-FS-ILS that carry local search invocation based on the VNS. It maintains two data structures which are *iScore* and *pScore*. The *iScore* is for storing the level of influence of a particular local search heuristic to produce the best new solutions. The *pScore* is for measuring the effectiveness of applying two local search heuristics in succession. The input to the procedure is a perturbed solution that goes through hill-climbing intensification and the best solution found at the end of the procedure is returned. $S_{pert}$, $S_{bl}$, and $S^*$ respectively represent the perturbed solution, the solution produced after applying a local search heuristic, and the best solution found so far during the intensification stage. The roulette wheel scheme was employed to select the first local search heuristic before the while loop using the vector iScore. However, within the loop, the next local search heuristic to apply is based on two parameters: the matrix pScore and the previously applied local search heuristic captured by its index *prev*. It is important to note that all the entries of the two data structures are initially set to 1.

---

**Algorithm 8:** LS-Seq procedure

---

**Input:** $S_{pert}$, the solution from the perturbation stage

$S_{bl} \leftarrow S_{pert}$

$cur \leftarrow RWS(iScore)$ /*Based on iScore, select the index of the current LLH*/

$h_l \leftarrow L_H[cur]$        /*get HyFlex id of the LLH*/

$S^* \leftarrow apply(h_l, S_{bl})$  /*apply the LLH*/

**while** $e(S^*) < e(S_{bl})$
   $S_{bl} \leftarrow S^*$
   $prev \leftarrow cur$/*update the index of the previous local search LLH*/
   $cur \leftarrow RWS(pScore, prev)$/*select the index of the next LLH*/
   $h_l \leftarrow L_H[cur]$
   $S^* \leftarrow apply(h_l, S_{bl})$
**end while**
**Output:** $S_{bl}$, the best solution produced from the LS-Seq procedure

Figure 2 illustrates how the data structures of LS-Seq were updated. In the first view, heuristic L0 has only been involved once during the update of the best global solution. L1 and L2 have been involved two times and five times, respectively, hence the vector entries 2, 3, 6. During the call of the LS-Seq procedure, L0, L1, and L2 are applied in the given order, leading to the update of both the iScore vector and the pScore matrix. Specifically, the intersection of L0–L1 was updated to 3 while that of L1–L2 was updated to 7 to strengthen the selection of L1 after the application of L0 and the selection of L2 after the application of L1. Since all the local search heuristics were applied once based on the sequence L0, L1, L2, they all received an increment of 1 in their respective iScore entries to change {2, 3, 6} to {3, 4, 7}. At the end of the third view, only L2 is applied during the current call of LS-Seq, and its iScore value is increased to 9. The iScore vector has the entries {3, 5, 9}, while no update is performed on the pScore matrix because the sequence {L2} is a singleton. Eventually, the local search invocation almost creates a sequence based on the inter-neighborhood strengths discovered over time.

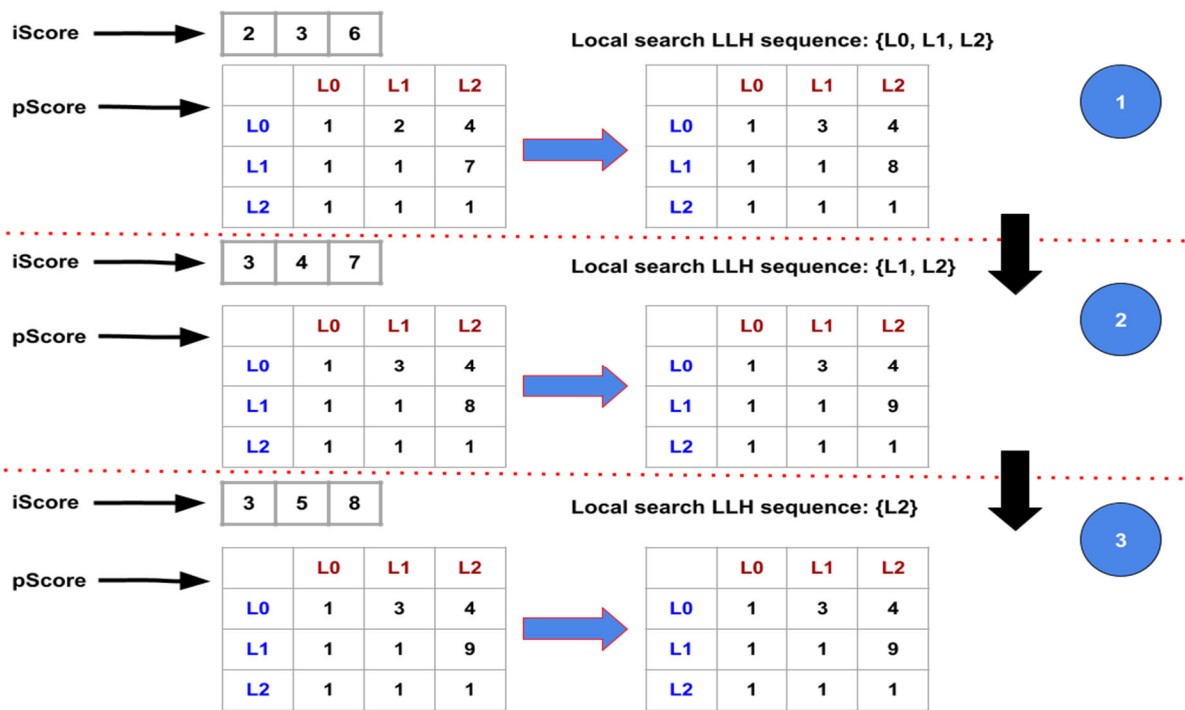

**Figure 2.** Illustration of LS-Seq update procedure.

## 4. Experimental Results

The EA-ILS hyper-heuristic has been tested on all instances of each problem in the extended HyFlex suite. The stopping criterion for the EA-ILS hyper-heuristic was set according to the execution time returned by a program on the machine used for the experimentation. The time limit for the EA-ILS hyper-heuristic on the machine used for an experiment is 530 s, which is the equivalent of 600 s on a standard testing computer

according to the CHeSC 2011 organizers. The performance of the EA-ILS hyper-heuristic on the extended HyFlex domains was benchmarked against those of other hyper-heuristics that have been tested on the domains. The other competing algorithms include FS-ILS, NR-FS-ILS, AdapHH, EPH, SR-IE, SR-AM, and SSHH. Both SR-IE and SR-AM select LLHs at random but differ in their acceptance mechanisms. SR-IE accepts solutions with equal or better quality than the incumbent solution while SR-AM accepts every proposed solution regardless of its quality. The data generated for the performances of the three domains were obtained from the webpage (https://github.com/Steven-Adriaensen/hyflext accessed on 10 September 2022), which the exception of that of the SSHH. The median objective function values (ofvs) of the solutions obtained for the SSHH can be found in a paper [25]. Tables 3 and 4 respectively highlight the performance of the EA-ILS hyper-heuristic in terms of its overall best and median best ofvs obtained across the 30 instances of the problem domains in comparison with the top six performing hyper-heuristics reported in [25], where the values in bold style denote the best values. In Table 3, the percentage deviation in the best ofv of the EA-ILS hyper-heuristic in each instance from the best-known values reported in [24], denoted by $\Delta$(%), is presented.

**Table 3.** Best ofvs Obtained by the EA-ILS in Comparison to the Existing Hyper-Heuristics.

| Domain | I | $\Delta$(%) | EA-ILS | AdapHH | FS-ILS | NR-FS-ILS | EPH | SSHH | SR-AM |
|---|---|---|---|---|---|---|---|---|---|
| Knapsack Problem | 0 | 0.0000 | **−104,046** | **−104,046** | **−104,046** | **−104,046** | **−104,046** | **−104,046** | **−104,046** |
| | 1 | 0.0026 | **−1,263,828** | −1,263,317 | −1,238,256 | −1,251,478 | −1,257,833 | −1,261,320 | −1,218,285 |
| | 2 | 0.0592 | **−243,001** | −242,841 | −239,378 | −241,794 | −242,198 | −242,963 | −239,346 |
| | 3 | 0.0009 | −431,359 | **−431,363** | −431,347 | −431,354 | −431,350 | −431,362 | −431,330 |
| | 4 | 0.0000 | **−396,167** | **−396,167** | **−396,167** | **−396,167** | **−396,167** | **−396,167** | **−396,167** |
| | 5 | 1.8119 | −4,337,691 | **−4,378,410** | −4,266,654 | −4,248,962 | −4,341,328 | −4,268,665 | −4,264,094 |
| | 6 | 0.7982 | **−946,555** | −943,371 | −938,125 | −938,646 | −943,247 | −943,136 | −934,838 |
| | 7 | 0.0000 | **−1,577,175** | **−1,577,175** | −1577,166 | −1,577,166 | **−1,577,175** | **−1,577,175** | **−1,577,175** |
| | 8 | 0.0014 | **−1,530,515** | −1,530,497 | −1530,479 | −1,530,480 | −1,530,514 | −1,530,511 | −1,530,476 |
| | 9 | 0.0063 | −1,467,362 | −1,467,362 | −1467,357 | −1,467,353 | **−1,467,387** | −1,467,362 | −1,467,357 |
| Quadratic Assignment Problem | 0 | 0.0000 | **152,002** | 152,046 | **152,002** | 152,044 | 152,116 | 152,224 | 152,280 |
| | 1 | 0.0065 | 153,900 | **153,890** | 153,916 | **153,890** | 153,942 | 154,130 | 154,160 |
| | 2 | 0.0000 | **147,862** | 147,868 | 147,898 | 147,866 | 147,872 | 147,930 | 148,058 |
| | 3 | 0.0013 | **149,578** | 149,672 | 149,596 | 149,594 | 149,762 | 149,782 | 149,846 |
| | 4 | 0.7836 | **21,217,438** | 21,303,448 | 21,246,800 | 21,242,104 | 21,279,308 | 21,325,030 | 21,454,914 |
| | 5 | 0.0000 | **1,185,996,137** | **1,185,996,137** | 1,186,007,112 | 1,186,055,449 | **1,185,996,137** | 1,186,663,179 | 1,187,672,220 |
| | 6 | 13.1320 | 499,802,038 | 500,066,316 | 499,728,427 | **499,571,734** | 500,645,098 | 500,015,697 | 499,912,219 |
| | 7 | 2.2737 | 44,846,660 | 44,825,454 | 44,840,214 | 44,843,206 | **44,817,780** | 44,855,568 | 44,850,886 |
| | 8 | 6.8364 | 8,141,608 | 8,148,152 | 8,152,748 | 8,147,252 | **8,140,772** | 8,151,040 | 8,154,234 |
| | 9 | 0.0022 | **273,044** | 273,054 | 273,112 | 273,054 | 273,276 | 273,216 | 273,262 |
| Maximum-cut Problem | 0 | 0.0000 | **−41,684,814** | **−41,684,814** | **−41,684,814** | **−41,684,814** | −41,446,603 | −41,517,765 | −40,699,212 |
| | 1 | 1.2953 | **−279,538,175** | −274,477,564 | −263,474,137 | −261,502,273 | −269,348,577 | −277,548,425 | −265,390,780 |
| | 2 | 0.0979 | −3061 | −3053 | −3057 | −3056 | −3041 | **−3062** | −3056 |
| | 3 | 0.0328 | −3049 | −3032 | −3033 | −3042 | −3017 | **−3050** | −3044 |
| | 4 | 0.2621 | −3044 | −3037 | −3037 | −3043 | −3017 | **−3051** | −3043 |
| | 5 | 0.8159 | −13,250 | −13,177 | −13,158 | −13,152 | −13,140 | **−13,300** | −13,230 |
| | 6 | 1.3006 | **−1366** | −1334 | −1318 | −1332 | −1272 | −1358 | −1332 |
| | 7 | 1.4856 | **−10,146** | −9929 | −9744 | −9765 | −9851 | −10,125 | −9951 |
| | 8 | 0.0000 | **−458** | **−458** | −456 | −456 | −440 | **−458** | −456 |
| | 9 | 2.3889 | −2942 | −2832 | −2718 | −2730 | −2760 | **−2960** | −2862 |

**Table 4.** Median ofvs Obtained by the EA-ILS in Comparison to the Existing Hyper-Heuristics.

| Domain | I | EA-ILS | AdapHH | FS-ILS | NR-FS-ILS | EPH | SSHH | SR-AM |
|---|---|---|---|---|---|---|---|---|
| Knapsack Problem | 0 | **−104,046** | **−104,046** | **−104,046** | **−104,046** | **−104,046** | **−104,046** | −104,025 |
| | 1 | **−1,262,154** | −1,258,634 | −1,220,103 | −1,231,767 | −1,253,074 | −1,247,642 | −1,209,914 |
| | 2 | **−242,603** | −242,104 | −236,813 | −239,578 | −240,663 | −241,934 | −238,397 |
| | 3 | −431,344 | **−431,351** | −431,297 | −431,312 | −431,333 | −431,350 | −431,311 |
| | 4 | **−396,167** | **−396,167** | −395,941 | −395,654 | **−396,167** | **−396,167** | **−396,167** |
| | 5 | −4,256,586 | **−4,328,770** | −3,756,992 | −3,697,266 | −4,283,926 | −4,251,693 | −4,248,962 |
| | 6 | **−940,291** | −937,868 | −906,490 | −895,516 | −936,200 | −929,052 | −923,973 |
| | 7 | **−1,577,175** | **−1,577,175** | −1,572,999 | −1,572,999 | **−1,577,175** | **−1,577,175** | **−1,577,175** |
| | 8 | −1,530,477 | −1,530,463 | −1,347,297 | −1,346,608 | −1,530,471 | **−1,530,477** | −1,530,453 |
| | 9 | **−1,467,357** | −1,467,353 | −1,463,681 | −1,462,759 | **−1,467,357** | **−1,467,357** | −1,467,353 |
| Quadratic Assignment Problem | 0 | **152,102** | 152,214 | 152,196 | 152,196 | 152,388 | 152,572 | 152,402 |
| | 1 | **154,010** | 154,164 | 154,088 | 154,166 | 154,390 | 154,492 | 154,290 |
| | 2 | **147,890** | 147,970 | 148,002 | 147,978 | 148,122 | 148,374 | 148,190 |
| | 3 | **149,722** | 149,850 | 149,858 | 149,828 | 150,144 | 150,366 | 149,992 |
| | 4 | **21,306,194** | 21,366,688 | 21,309,208 | 21,321,554 | 21,401,254 | 21,419,490 | 21,518,130 |
| | 5 | **1,187,379,429** | 1,187,875,748 | 1,187,490,923 | 1,187,383,316 | 1,189,221,001 | 1,190,346,287 | 1,189,321,259 |
| | 6 | **501,094,667** | 502,937,700 | 503,088,738 | 502,654,006 | 502,409,100 | 504,406,437 | 502,293,807 |
| | 7 | 44,867,334 | **44,858,394** | 44,874,028 | 44,873,022 | 44,860,940 | 44,892,452 | 44,866,876 |
| | 8 | **8,152,360** | 8,163,764 | 8,169,250 | 8,162,592 | 8,163,304 | 8,179,752 | 8,168,990 |
| | 9 | **273,312** | 273,414 | 273,362 | 273,336 | 273,630 | 273,622 | 273,512 |
| Maximum-cut Problem | 0 | **−41,398,025** | −41,348,693 | −41,348,693 | −41,145,032 | −40,953,212 | −41,101,646 | −40,502,841 |
| | 1 | **−276,571,977** | −255,265,025 | −255,265,025 | −257,764,081 | −260,608,752 | −273,938,900 | −263,151,470 |
| | 2 | −3051 | −3044 | −3041 | −3044 | −3023 | **−3056** | −3046 |
| | 3 | −3035 | −3025 | −3020 | −3025 | −3004 | **−3040** | −3033 |
| | 4 | −3039 | −3026 | −3026 | −3028 | −3004 | **−3041** | −3035 |
| | 5 | −13,211 | −13,126 | −13,083 | −13,091 | −13,065 | **−13,243** | −13,177 |
| | 6 | **−1356** | −1314 | −1302 | −1304 | −1206 | −1352 | −1322 |
| | 7 | **−10,078** | −9823 | −9632 | −9668 | −9794 | −10,074 | −9878 |
| | 8 | **−456** | −450 | −450 | −450 | −430 | −454 | −454 |
| | 9 | −2902 | −2786 | −2676 | −2680 | −2,648 | **−2912** | −2814 |

## 4.1. Comparison of Hyper-Heuristic Algorithms

The F1 scoring system is one of the most popular metrics for evaluating hyper-heuristics [11]. The competing algorithms are assigned points based on the ofvs of their median (16th) best solutions obtained after 31 trials on each instance of a problem in the given test suite. Points 10, 8, 6, 5, 4, 3, 2, and 1 are awarded to the best hyper-heuristic down to the eighth best hyper-heuristic on the instance of a problem, respectively. Ties are handled by averaging the points that would have been given to the hyper-heuristics if there was no tie and assigning the average score to each of the hyper-heuristics. The results following the evaluation of the EA-ILS hyper-heuristic against the existing seven hyper-heuristics using the F1 ranking are presented in the form of bar charts in Figure 3. The EA-ILS hyper-heuristic emerged as the winner of the contest across the problem domains used as the basis for evaluation (Figure 3a–d). The scores of the EA-ILS hyper-heuristic on KP, QAP, and MAC problems are 78.2, 95.0, and 90.0, respectively, bearing in mind that the total obtainable score on each domain is 100.0. The top three performing hyper-heuristics are EA-ILS with 263.2 points, AdapHH with 180.2 points, and SSHH with 171.2 points while SR-IE finished last with 48.5 points. The no-restart version of the FS-ILS hyper-heuristic (NR-FS-ILS) outperformed the original version (FS-ILS) in all the domains while the SR-AM hyper-heuristic finished fourth based on the overall rankings.

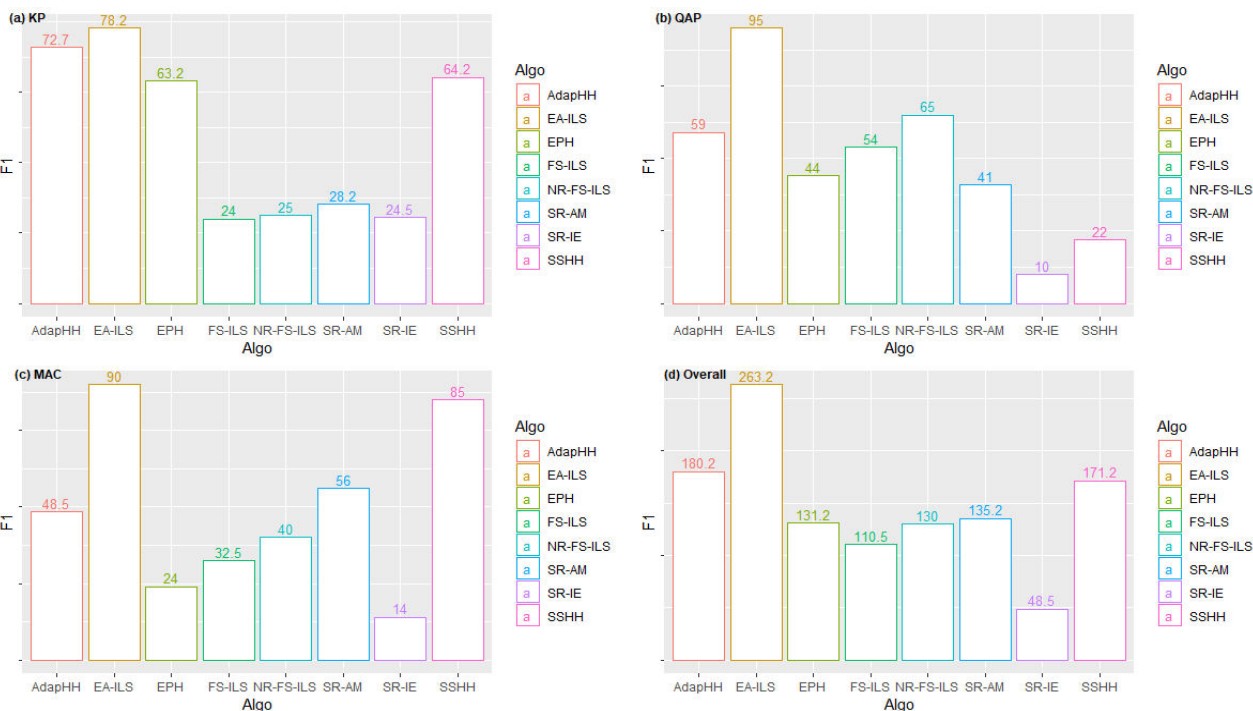

**Figure 3.** F1 scores of the hyper-heuristics on the three problem domains (**a**–**c**) and the sum of scores across the three domains (**d**).

The performance of the EA-ILS hyper-heuristic was further compared with those of six other hyper-heuristics using the evaluation metrics of μ-norm, μ-rank, best, and worst [24,25]. The μ-norm is an average normalized evaluation function and is a more robust evaluation metric than the F1 scoring system. This is because it can evaluate the performance of a hyper-heuristic based on the quality of the 31 solutions obtained over 31 trials on a problem instance relative to its competitors. The μ-rank is the average rank of the median cost obtained by each metric and is based on the value of the μ-norm. This means that the comparative hyper-heuristics are ranked based on the increasing value of μ-norm. The highest μ-rank is 1 while the lowest is n, where n is the number of competing hyper-heuristics. The metrics best and worst refer to the number of instances for which a hyper-heuristic obtained the best (highest) and worst (lowest) median ofvs. The SSHH algorithm is not included in the evaluation because the quality of the 31 solutions obtained from its test on the instances could not be obtained. Table 5 presents the AdapHH hyper-heuristic as the closest challenger to the EA-ILS hyper-heuristic on the knapsack problem, but the metric values for the EA-ILS hyper-heuristic still establish its superiority. In the other two problem domains, the EA-ILS hyper-heuristic was dominant over the comparative hyper-heuristics across evaluation metrics and problem domains as shown in Tables 5–8. The overall performance of the EA-ILS hyper-heuristic is seen in Table 8 to be better than those of the comparative hyper-heuristics across the evaluation metrics.

**Table 5.** Evaluation Results on the Knapsack Problem.

| Rank | Hyper-Heuristic | μ-Norm | μ-Rank | Best | Worst |
|---|---|---|---|---|---|
| 1 | EA-ILS | 0.0210 | 1.30 | 8 | 0 |
| 2 | AdapHH | 0.0302 | 1.70 | 5 | 0 |
| 3 | EPH | 0.0556 | 2.00 | 4 | 0 |
| 4 | SR-AM | 0.1507 | 4.00 | 2 | 0 |
| 5 | SR-IE | 0.3300 | 5.50 | 0 | 4 |
| 6 | NR-FS-ILS | 0.3628 | 5.30 | 1 | 6 |
| 7 | FS-ILS | 0.3967 | 5.40 | 1 | 2 |

**Table 6.** Evaluation Results on the Quadratic Assignment Problem.

| Rank | Hyper-Heuristic | μ-Norm | μ-Rank | Best | Worst |
|------|-----------------|--------|--------|------|-------|
| 1 | EA-ILS | 0.0727 | 1.30 | 9 | 0 |
| 2 | NR-FS-ILS | 0.1036 | 2.90 | 0 | 0 |
| 3 | AdapHH | 0.1063 | 3.40 | 1 | 0 |
| 4 | FS-ILS | 0.1071 | 3.80 | 0 | 0 |
| 5 | EPH | 0.1369 | 4.60 | 0 | 0 |
| 6 | SR-AM | 0.1486 | 4.90 | 0 | 0 |
| 7 | SR-IE | 0.6355 | 7.00 | 0 | 10 |

**Table 7.** Evaluation Results on the Maximum-Cut Problem.

| Rank | Hyper-Heuristic | μ-Norm | μ-Rank | Best | Worst |
|------|-----------------|--------|--------|------|-------|
| 1 | EA-ILS | 0.0886 | 1.00 | 10 | 0 |
| 2 | SR-AM | 0.2392 | 2.50 | 0 | 0 |
| 3 | AdapHH | 0.2658 | 3.00 | 0 | 0 |
| 4 | NR-FS-ILS | 0.3634 | 3.90 | 0 | 0 |
| 5 | FS-ILS | 0.3811 | 4.80 | 0 | 2 |
| 6 | EPH | 0.5116 | 5.60 | 0 | 1 |
| 7 | SR-IE | 0.7305 | 6.60 | 0 | 7 |

**Table 8.** Overall Performance on the Extended Hyflex Test Suite.

| Rank | Hyper-Heuristic | μ-Norm | μ-Rank | Best | Worst |
|------|-----------------|--------|--------|------|-------|
| 1 | EA-ILS | 0.0608 | 1.20 | 27 | 0 |
| 2 | AdapHH | 0.1341 | 2.70 | 6 | 0 |
| 3 | SR-AM | 0.1795 | 3.80 | 2 | 0 |
| 4 | EPH | 0.2347 | 4.07 | 4 | 1 |
| 5 | NR-FS-ILS | 0.2766 | 4.03 | 1 | 6 |
| 6 | FS-ILS | 0.2950 | 4.67 | 1 | 4 |
| 7 | SR-IE | 0.5653 | 6.37 | 0 | 21 |

*4.2. Statistical Significance of Hyper-Heuristic Algorithms*

The statistical significance of the superiority of the EA-ILS hyper-heuristic over the comparative hyper-heuristics was established across the three problem domains. The Friedman test was first performed on the median ofvs of the competing hyper-heuristics to statistically evaluate these values. The boxplot visualizations of the median ofvs per problem domain and overall performance are subsequently presented.

4.2.1. Friedman Test

The median ofvs obtained by the hyper-heuristics, including the SSHH, were subjected to the Friedman test at a significance level of 0.05. The test result returned a *p*-value less than $2.2 \times 10^{-16}$, which is an insignificant value. This result means that there is a significant difference between the median ofvs obtained by the EA-ILS hyper-heuristic across the three problem domains. The Friedman rankings are presented in Table 9 and indicate that the EA-ILS hyper-heuristic is the best with a rank of 1.67 while AdapHH and SSHH finished second and third, respectively. The two variants of ILS, which are FS-ILS and NR-FS-ILS, finished sixth and seventh, respectively.

**Table 9.** Ranking of the Hyper-Heuristics Using the Friedman Test.

| S/N | Hyper-Heuristic | Rank |
|-----|-----------------|------|
| 1 | EA-ILS | 1.67 |
| 2 | AdapHH | 3.38 |
| 3 | SSHH | 3.88 |
| 4 | SR-AM | 4.58 |
| 5 | EPH | 4.78 |
| 6 | NR-FS-ILS | 4.83 |
| 7 | FS-ILS | 5.45 |
| 8 | SR-IE | 7.35 |

4.2.2. Boxplot Analysis

The boxplots in Figure 4a–d represent the performance of the eight hyper-heuristics presented in Table 9. The ten median ofvs obtained by the eight hyper-heuristics over the ten instances of the problem domains were used as data points for plotting the boxplots. The overall boxplot in Figure 4d combines the data points from each domain to visualize the overall performance of each hyper-heuristic. The median ofv obtained by each algorithm in an instance i was normalized to a value in the range [0, 1] using Equation (1) to obtain a uniform evaluation $ofv(O, i, h, b, w)$ as follows:

$$ofv(O, i, h, b, w) = \frac{O_i^h - O_i^b}{O_i^w - O_i^b} \tag{1}$$

where $O_i^h$ is the median ofv obtained by a given hyper-heuristic h on an instance $i$, $O_i^b$ is the median ofv obtained by the best hyper-heuristic $b$ on an instance $i$, and $O_i^w$ is the median ofv obtained by the worst hyper-heuristic $w$ on an instance $i$. The best hyper-heuristic in an instance receives a value of 0.0 while the worst receives a value of 1.0 according to Equation (1). Figure 4 shows that the EA-ILS hyper-heuristic is the best performing across the problem domains.

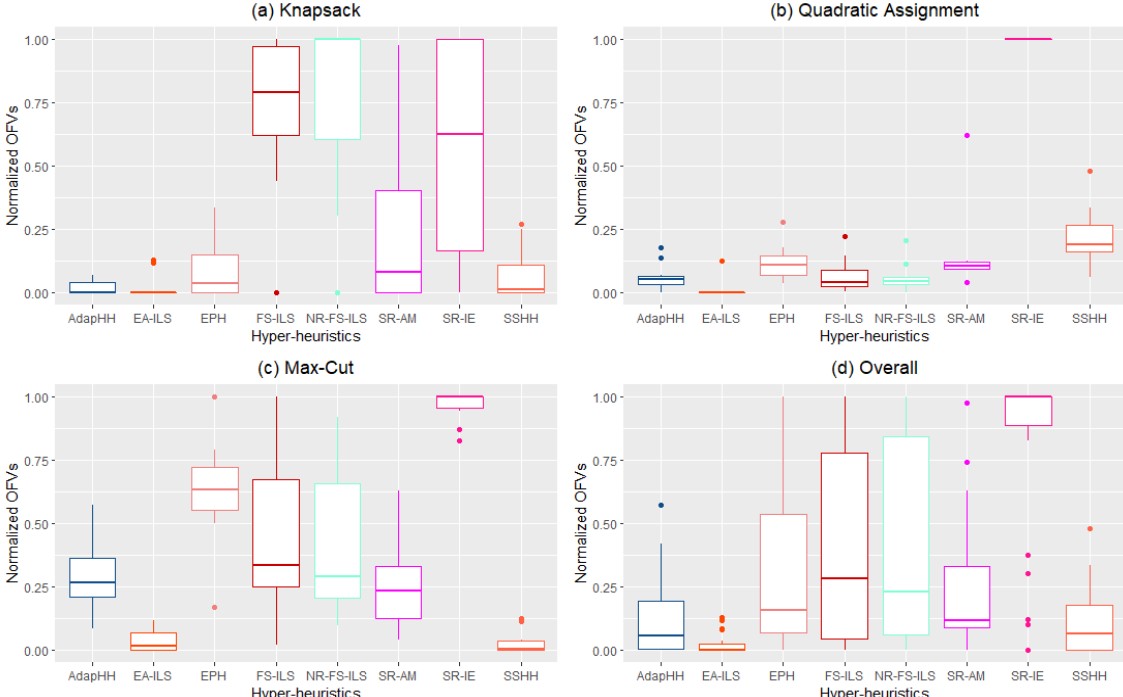

**Figure 4.** Boxplot visualizations of the median objective function values obtained by hyper-heuristics per domain (**a**–**c**) and the overall boxplot over all three domains (**d**).

*4.3. Application of EA-ILS Hyper-Heuristic to HyFlex Version 1.0*

The EA-ILS hyper-heuristic was tested on six problem domains of the first version of HyFlex (HyFlex V1.0) to further demonstrate its generality. The problem domains are Boolean satisfiability (SAT), bin packing (BP), personnel scheduling (PS), permutation flow-shop (PFS), traveling salesman problem (TSP), and vehicle routing problem (VRP). Tables 10 and 11 respectively show comparisons of the best and median ofvs of the EA-ILS hyper-heuristic obtained from its 31 runs in each instance against the state-of-the-art hyper-heuristics AdapHH, FS-ILS, and TS-ILS [48]. The values presented in the tables further establish the competitiveness of the EA-ILS hyper-heuristic when compared to the state-of-the-art ones. It is important to note that only algorithms that were run with the updated version of HyFlex v1.0 can be directly compared with the EA-ILS hyper-heuristic on the PS domain.

**Table 10.** Best ofvs Obtained by EA-ILS against three state-of-the-art Hyper-Heuristics.

| Domain | Instance | EA-ILS | AdapHH | FS-ILS | TS-ILS |
|---|---|---|---|---|---|
| MAX-SAT | SAT3 | 1.0 | 1.0 | 1.0 | **0.0** |
| | SAT5 | **1.0** | **3.0** | **1.0** | **1.0** |
| | SAT4 | **0.0** | **1.0** | **0.0** | **0.0** |
| | SAT10 | **1.0** | **1.0** | **1.0** | **1.0** |
| | SAT11 | **7.0** | **7.0** | **7.0** | **7.0** |
| Bin Pack-ing | BP7 | **0.01107109** | 0.0131 | 0.01384737 | 0.01569294 |
| | BP1 | 0.00339113 | **0.0028** | 0.00666972 | 0.00306951 |
| | BP9 | 0.00162385 | **0.0004** | 0.01004134 | 0.00049110 |
| | BP10 | 0.10829805 | 0.1083 | 0.10833606 | 0.10827981 |
| | BP11 | 0.00431172 | 0.0031 | 0.01047079 | **0.00115128** |
| Personnel Scheduling | PS5 | **15.0** | - | 17.0 | **15.0** |
| | PS9 | **9176.0** | - | 9486.0 | 9291.0 |
| | PS8 | **3136.0** | - | 3148.0 | 3142.0 |
| | PS10 | 1380.0 | - | **1360.0** | 1453.0 |
| | PS11 | **305.0** | - | 325.0 | 315.0 |
| Flowshop | PFS1 | **6210.0** | 6214.0 | 6214.0 | **6210.0** |
| | PFS8 | **26700.0** | 26757.0 | 26743.0 | 26744.0 |
| | PFS3 | **6303.0** | **6303.0** | **6303.0** | **6303.0** |
| | PFS10 | **11308.0** | 11318.0 | 11332.0 | **11308.0** |
| | PFS11 | **26511.0** | 26541.0 | 26547.0 | 26516.0 |
| Travelling Salesman | TSP0 | **48194.9** | **48194.9** | **48194.9** | **48194.9** |
| | TSP8 | 20732537.2 | 20752853.8 | 20933386.7 | **20662037.2** |
| | TSP2 | 6798.8 | 6797.5 | **6796.5** | 6798.6 |
| | TSP7 | 66017.2 | 66277.1 | 65748.4 | **65592.7** |
| | TSP6 | 52545.2 | 52383.8 | 52385.5 | **52308.7** |
| Vehicle Routing | VRP6 | 61943.4 | **58052.1** | 63429.7 | 60145.1 |
| | VRP2 | 12270.1 | 13304.9 | 12277.1 | **12266.9** |
| | VRP5 | 143902.4 | 145481.5 | **142481.3** | 142607.9 |
| | VRP1 | 20652.2 | 20652.3 | **20651.6** | 20652.2 |
| | VRP9 | 144030.2 | 146154.1 | 144686.3 | **143479.0** |

**Table 11.** Median ofvs Obtained by EA-ILS against three state-of-the-art Hyper-Heuristics.

| Domain | Instance | EA-ILS | AdapHH | FS-ILS | TS-ILS |
|---|---|---|---|---|---|
| MAX-SAT | SAT3 | 4.0 | 3.0 | **2.0** | **2.0** |
| | SAT5 | 5.0 | 5.0 | **3.0** | **3.0** |
| | SAT4 | 2.0 | 2.0 | **1.0** | **1.0** |
| | SAT10 | 6.0 | 3.0 | 2.0 | **1.0** |
| | SAT11 | 9.0 | **8.0** | **8.0** | **8.0** |

| | | | | | |
|---|---|---|---|---|---|
| | BP7 | 0.01612590 | **0.01607535** | 0.01851932 | 0.01876799 |
| | BP1 | 0.00354067 | 0.00360372 | 0.00751493 | **0.00350695** |
| Bin Packing | BP9 | 0.00267695 | 0.00356587 | 0.01122520 | **0.00052035** |
| | BP10 | 0.10831818 | **0.10828303** | 0.10840170 | 0.10828402 |
| | BP11 | 0.00755952 | 0.00354259 | 0.01355520 | **0.00142866** |
| | PS5 | **20.0** | - | 23.0 | 21.0 |
| | PS9 | 9560.0 | - | 9763.0 | **9548.0** |
| Personnel Scheduling | PS8 | **3164.0** | - | 3236.0 | 3181.0 |
| | PS10 | **1550.0** | - | 1635.0 | **1550.0** |
| | PS11 | **325.0** | - | 345.0 | 330.0 |
| | PFS1 | **6224.0** | 6240.0 | 6241.0 | 6232.0 |
| | PFS8 | **26769.0** | 26814.0 | 26797.0 | 26785.0 |
| Flowshop | PFS3 | **6323.0** | 6326.0 | **6323.0** | 6325.0 |
| | PFS10 | 11344.0 | 11359.0 | 11374.0 | **11340.0** |
| | PFS11 | **26583.0** | 26643.0 | 26605.0 | 26601.0 |
| | TSP0 | **48194.9** | **48194.9** | **48194.9** | **48194.9** |
| | TSP8 | 21333200.2 | 20822145.6 | 21172591.7 | **20779493.2** |
| Travelling Salesman | TSP2 | 6805.8 | 6810.5 | 6806.7 | **6805.3** |
| | TSP7 | 66483.8 | 66879.8 | 66415.3 | **66133.0** |
| | TSP6 | 53997.5 | 53099.8 | 52840.8 | 53762.4 |
| | VRP6 | 66178.5 | **60900.6** | 65638.8 | 63709.0 |
| | VRP2 | 13286.3 | 13347.6 | **12308.0** | 13292.8 |
| Vehicle Routing | VRP5 | 146813.7 | 148516.8 | 146871.0 | 145401.5 |
| | VRP1 | **20654.1** | 20656.6 | **20654.1** | 20654.7 |
| | VRP9 | 145765.2 | 148689.2 | 146242.7 | **145205.4** |

The entries for TS-ILS and FS-ILS were taken from a previous study [48] where both methods were tested on the same machine. The EA-ILS hyper-heuristic comfortably outperformed others on the PS and PFS problem domains. Moreover, it has outperformed the FS-ILS hyper-heuristic on the BP problem but is inferior to others on the SAT according to the median ofvs obtained. The performances of the 20 CHeSC entries, EA-ILS, TS-ILS, and FS-ILS were further analyzed using the F1 ranking test. Table 12 shows the results of the EA-ILS hyper-heuristic versus only the 20 CHeSC entries while Figure 5 shows the results of the EA-ILS hyper-heuristic vs. the 20 CHeSC entries, FS-ILS, and TS-ILS. The overall F1 ranking of the EA-ILS hyper-heuristic based on the total number of algorithms is 3 as observed in Figure 5, behind the TS-ILS and FS-ILS hyper-heuristics. We recorded a total score of 126.25 and a deficit of 4.50 for the score of the FS-ILS hyper-heuristic. It finished fourth, third, first, fifth, and joint third in the SAT, BP, PFS, TSP, and VRP problem domains, respectively. The results of the PS of the EA-ILS, FS-ILS, and TS-ILS hyper-heuristics could not be used for these rankings. This is because of the use of the updated HyFlex library, which was released to correct the errors identified on the PS domain.

**Table 12.** F1 scores of top 7 Hyper-heuristics after comparing EA-ILS with the 20 CHeSC entries.

| | Problem Domains | | | | | |
|---|---|---|---|---|---|---|
| **Hyper-Heuristic** | **SAT** | **BP** | **PFS** | **TSP** | **VRP** | **Overall** |
| EA-ILS | 21.2 | 36.0 | 48.0 | 24.0 | 29.5 | 158.7 |
| AdapHH | 33.6 | 42.0 | 30.0 | 36.25 | 13.0 | 154.85 |
| ML | 11.0 | 8.0 | 31.5 | 11.0 | 19.5 | 81.0 |
| VNS-TW | 33.6 | 2.0 | 26.0 | 15.25 | 4.0 | 80.85 |
| PHUNTER | 8.0 | 2.0 | 6.0 | 24.25 | 30.0 | 70.25 |
| EPH | 0.0 | 6.0 | 16.0 | 32.25 | 11.0 | 65.25 |

| | NAHH | 11.5 | 18.0 | 18.5 | 11.0 | 5.0 | 64.0 |

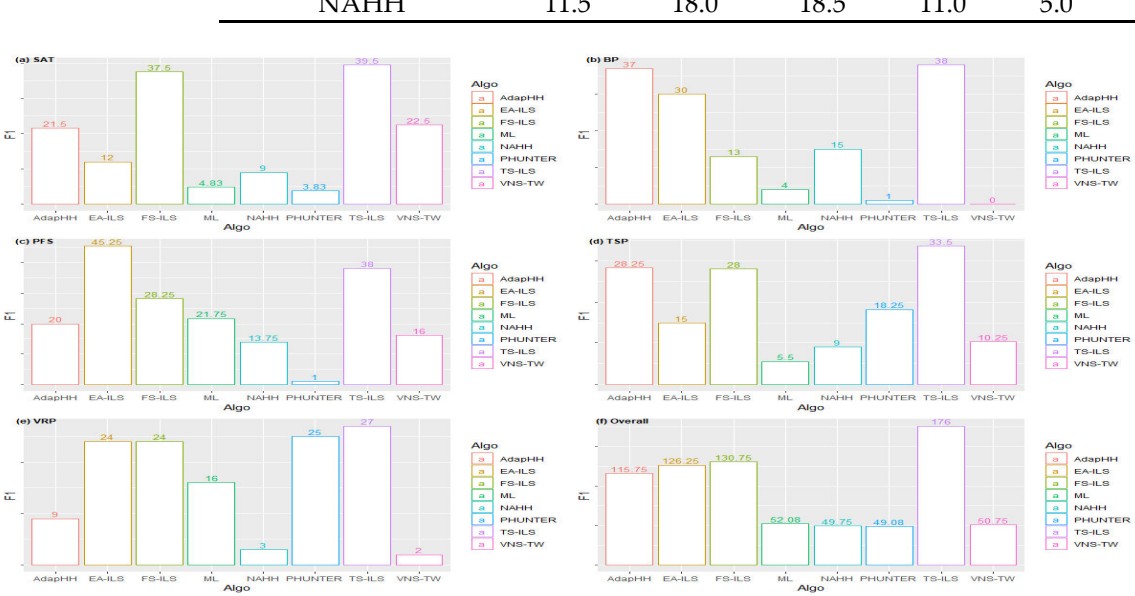

**Figure 5.** F1 plots of EA-ILS with other top hyper-heuristics on the HyFlex domains.

The comparison of three ILS-based hyper-heuristics using a more robust $\mu$-norm performance metric [24,25] was performed across the six domains from HyFlex v1.0. The performance metric normalizes the 31 objective function values obtained by a given hyperheuristic, thus using the results of all 31 trials per instance as a basis for the evaluation. A fair comparison is possible based on the six domains of HyFlex v1.0 because the results for the PS achieved by the FS-ILS hyper-heuristic and the TS-ILS hyper-heuristic were based on the updated library for the domain. Table 13 summarizes the benchmarking of the performance of the EA-ILS hyper-heuristic with the results obtained by the FS-ILS and TS-ILS hyper-heuristics on the original six problem domains.

**Table 13.** Benchmarking EA-ILS with FS-ILS and TSILS on the six domains of HyFlex v1.0 using the $\mu$-norm metric

| Hyper-Heuristic | Problem Domains | | | | | | |
| | SAT | BP | PS | PFS | TSP | VRP | Overall |
|---|---|---|---|---|---|---|---|
| TS-ILS | **0.1341** | **0.1052** | 0.3735 | 0.4515 | 0.2852 | **0.3514** | **0.2835** |
| EA-ILS | 0.3677 | 0.1779 | **0.3418** | **0.3964** | 0.4435 | 0.3982 | 0.3542 |
| FS-ILS | 0.1365 | 0.5547 | 0.5505 | 0.5601 | **0.2692** | 0.3653 | 0.4061 |

It is evident from Table 13 that the EA-ILS hyper-heuristic is highly competitive, even in the six domains of HyFlex v1.0. Its weaknesses are evidenced by its application to the SAT and TSP problem domains. Although Table 10 suggests that the hyper-heuristic can find high-quality solutions, it is perhaps the lack of consistency in evolving the most effective perturbative heuristics, especially on the SAT domain, that has made it inferior to the two state-of-the-art hyper-heuristics. However, the EA-ILS hyper-heuristic outperformed both the FS-ILS and TS-ILS hyper-heuristics in the PS and PFS problem domains. This is highly commendable because of the strong performances of both competitors in the PFS domain and the strong performance of TS-ILS in the PS domain. The EA-ILS hyper-heuristic was not far from the best method for BP and VRP problems. Overall, it had a better $\mu$-norm score than the FS-ILS hyper-heuristic, suggesting its better generalization ability across the original six problem domains.

## 4.4. Effect of Local Search Procedure

The local search procedure (LS-Seq) of the EA-ILS hyper-heuristic was compared with the VND-style local search, tagged LS-VND, commonly used in the FS-ILS hyper-heuristic and NR-FS-ILS. The purpose was to experimentally investigate the effect of the local search component of the EA-ILS hyper-heuristic on its performance. Two representative instances were selected for each problem domain for experimentation. The control hyper-heuristic was constructed by replacing LS-Seq with LS-VND in the EA-ILS hyper-heuristic. The comparative results are presented in Table 14, wherein a shaded cell means the set-up has a better objective function value than the alternative set-up in the adjacent cell. The $\alpha$ and $\beta$ set-ups were run in an overlapped manner such that when a single run or trial is completed on a sample instance for a set-up, the run of the next set-up on the same instance immediately follows. Moreover, both set-ups start with the same random number seed for each run throughout the entirety of the sample instances. This mechanism ensures fairness in the experiment. The meaning of the column names in Table 14 along with other important information are described as follows: The column name $\alpha$ represents the EA-ILS hyper-heuristic with the LS-Seq procedure as the intensification module while $\beta$ is the control hyper-heuristic with the LS-VND procedure as the intensification module. $\alpha\_iter$ is the number of iterations covered by $\alpha$ while solving a problem instance during a run. $\beta\_iter$ is the number of iterations covered by $\beta$ while solving a problem instance during a run. $\alpha\_lsi$ is the average number of calls to local search heuristics per iteration by $\alpha$. $\beta\_lsi$ is the average number of calls to local search heuristics per iteration by $\beta$. The numbers of local search heuristics on the KP, QAP, and MAC domains are 6, 2, and 3, respectively.

**Table 14.** Test Results from the Overlapping Runs of the two EA-ILS Set-ups.

|  | $\alpha$ | $\beta$ | $\alpha_{iter}$ | $\beta_{iter}$ | $\alpha_{lsi}$ | $\beta_{lsi}$ |
|---|---|---|---|---|---|---|
| KP1-1 | −1,262,437 | −1,255,857 | 1561 | 414 | 6.52 | 31.09 |
| KP1-2 | −1,260,453 | −1,256,773 | 1234 | 368 | 5.95 | 35.49 |
| KP1-3 | −1,262,433 | −1,259,963 | 1655 | 356 | 7.00 | 39.09 |
| KP1-4 | −1,263,828 | −1,256,390 | 1929 | 385 | 9.87 | 34.76 |
| KP1-5 | −1,254,789 | −1,259,903 | 2089 | 382 | 5.07 | 34.47 |
| KP5-1 | −4,276,582 | −4,258,774 | 182 | 15 | 10.13 | 114.20 |
| KP5-2 | −4,258,106 | −3,974,568 | 145 | 16 | 3.16 | 111.25 |
| KP5-3 | −4,251,970 | −4,248,962 | 100 | 7 | 14.17 | 199.14 |
| KP5-4 | −4,259,539 | −3,683,130 | 158 | 8 | 9. 53 | 203. 50 |
| KP5-5 | −4,270,759 | −4,248,962 | 177 | 9 | 6.06 | 162.89 |
| QAP0-1 | 152,068 | 152,164 | 1876 | 2273 | 1.89 | 2.84 |
| QAP0-2 | 152,360 | 152,026 | 1594 | 2928 | 1.80 | 2.92 |
| QAP0-3 | 152,398 | 152,070 | 2408 | 2234 | 1.83 | 2.87 |
| QAP0-4 | 152,076 | 152,086 | 1895 | 2559 | 1.82 | 2.83 |
| QAP0-5 | 152,048 | 152,060 | 3466 | 2099 | 1.83 | 2.81 |
| QAP7-1 | 44,884,414 | 44,870,046 | 68 | 81 | 1.78 | 2.79 |
| QAP7-2 | 44,866,080 | 44,882,160 | 180 | 156 | 1.90 | 2.96 |
| QAP7-3 | 44,859,006 | 44,873,662 | 164 | 187 | 1.86 | 2.84 |
| QAP7-4 | 44,870,856 | 44,854,654 | 131 | 233 | 1.79 | 2.83 |
| QAP7-5 | 44,879,374 | 44,843,808 | 180 | 251 | 1.89 | 2.87 |
| MAC5-1 | −13,217 | −13,172 | 10,718 | 6931 | 2.17 | 4.25 |
| MAC5-2 | −13,226 | −13,139 | 13,920 | 6733 | 2.27 | 4.28 |
| MAC5-3 | −13,193 | −13,246 | 10,297 | 6740 | 2.23 | 4.36 |
| MAC5-4 | −13,227 | −13,214 | 10,923 | 6696 | 2.25 | 4.38 |
| MAC5-5 | −13,186 | −13,166 | 10,494 | 6732 | 2.27 | 4.33 |
| MAC6-1 | −1358 | −1354 | 38,313 | 23,406 | 1.91 | 4.19 |

| | | | | | |
|---|---|---|---|---|---|
| MAC6-2 | −1352 | −1358 | 38,719 | 22,350 | 2.05 | 4.29 |
| MAC6-3 | −1362 | −1344 | 38,209 | 23,286 | 1.92 | 4.25 |
| MAC6-4 | −1350 | −1352 | 49,983 | 24,363 | 1.86 | 4.18 |
| MAC6-5 | −1356 | −1350 | 33,797 | 26,373 | 2.04 | 4.04 |

The ofv of each run from the two set-ups were normalized to values in the range [0, 1] using the maximum and minimum ofvs over five trials completed for each sample instance to perform the Wilcoxon signed rank sum test. The reported $p$-value was 0.2249 at a 0.05 level of significance. The obtained $p$-value indicates that across the entire set of 30 data points, there is no significant difference between the performances of the two experimental set-ups of the EA-ILS hyper-heuristic. This finding can be attributed to the performance of the EA-ILS hyper-heuristic with the VND style of local search in the QAP domain. It was highly competitive or even slightly better than the EA-ILS hyper-heuristic with LS-Seq in the two instances of the QAP domain. In fact, after the removal of 10 data points that corresponded to the performance in QAP0 and QAP7 instances, a $p$-value of **0.01962** was returned from the Wilcoxon signed rank sum test. This would suggest a significant difference in the performances of the two set-ups.

The sum of ranks for the entire set of data points for which the EA-ILS hyper-heuristic with LS-Seq outperformed the control algorithm of the EA-ILS hyper-heuristic with LS-VND is 291 while the sum of ranks of the opposite outcome is 173. A careful observation of the $\alpha\_iter$ and $\beta\_iter$ values indicates that the first set-up completes a better diversification–intensification cycle than the control set-up in the KP instances. Peradventure this is the reason for the huge difference in the performances of the two set-ups. The LS-Seq set-up was able to visit many search areas and had leverage to escape the local optima over the alternative set-up because of the invocation of local search heuristics. The local search calls per iteration did not pose a challenge for the LS-VND set-up in the QAP domain based on its performance relative to that of LS-Seq. This observation may be attributed to the low number of local search heuristics available for the problem domain. Consequently, extensive local search calls improved its search capabilities and made it more competitive in the QAP domain. The other interesting phenomenon observed for the QAP instances is that the number of cycles completed by LS-Seq for a trial may not be higher than that of LS-VND. The EA-ILS hyper-heuristic with LS-Seq performed better than its LS-VND counterpart in the MAC instances. This can also be directly attributed to the average number of cycles completed by the former set-up in both the MAC5 and MAC6 instances. It can be safely concluded that the LS-Seq procedure performs faster with better search than LS-VND in both KP and MAC domains while the same statement cannot be made for the QAP domain.

The better performance obtained by the EA-ILS hyper-heuristic with LS-Seq in KP and MAC instances can be directly linked to the set-up covering many more iterations than its counterpart. Furthermore, experimental results suggest that the more local search heuristics that are available for a domain, the more likely an ILS with an LS-VND intensification procedure is to perform worse, compared to the ILS with a quicker escape from the intensification procedure such as LS-Seq. The LS-Seq has more advantages over the LS-VND in terms of its search capability. Perhaps if the number of local search heuristics for the QAP is increased to four, the LS-Seq outperforms the LS-VND in the same way it occurred in the KP and MAC instances. If a recommendation is to be made based on the experimental results, it is safe to use the VND style of local search strategy if the number of local search heuristics is small. The moment the number is at least four, a smarter local search procedure is required. This experiment has been able to show that the LS-VND procedure could impede the performance of the ILS, especially when the number of local search heuristics is large. This could be one of the main reasons that FS-ILS and NR-FS-ILS hyper-heuristics performed poorly in the KP and MAC problem domains.

### 4.5. Analysis of Effective Heuristics for EA-ILS Hyper-Heuristic

The distribution of effective perturbative heuristic sequences of the EA-ILS hyper-heuristic used to discover the best global solutions is provided in this section. Two sample instances were selected across the six problem domains to observe the low-level perturbative heuristics that make a positive impact on the search performance of the EA-ILS hyper-heuristic. The top 10 solutions produced by the EA-ILS hyper-heuristic were taken to analyze effective heuristics for a sample instance. The number of times a heuristic sequence updated the best solution was recorded for each sequence. The EA-ILS hyper-heuristic typically evolves many sequences during the optimization process and each of them may improve the best solution at least once. Therefore, the top **n** sequences with their improvement tallies were selected for the analysis. Figures 6–8 present the analysis of the twelve selected instances of the problem domains.

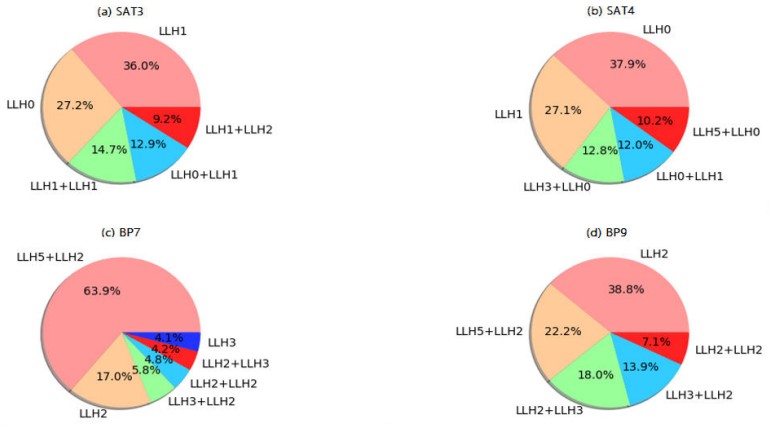

**Figure 6.** Analysis of two SAT and two BP instances.

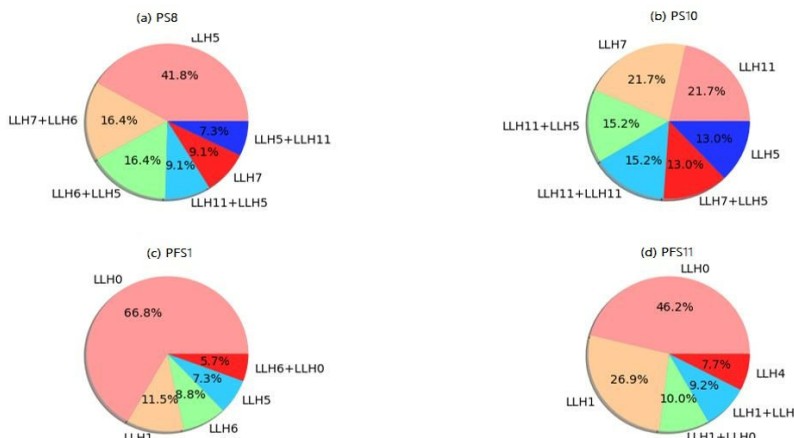

**Figure 7.** Analysis of two PS and two PFS instances.

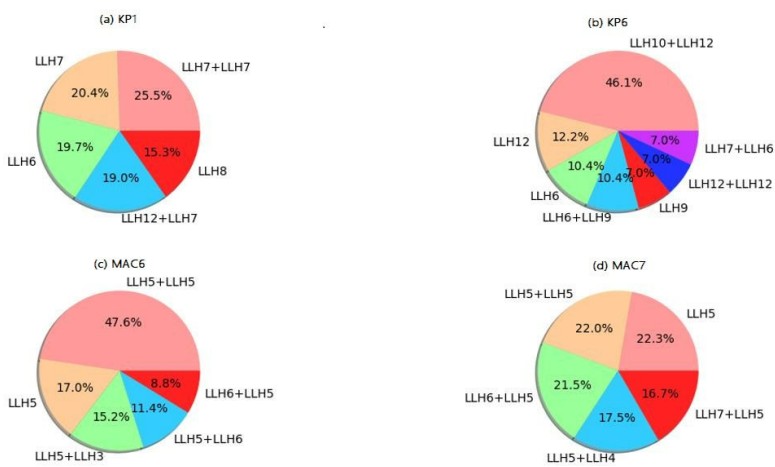

**Figure 8.** Analysis of two KP and two MAC instances.

More rewards were obtained for the SAT3 and SAT4 instances when a sole perturbative heuristic of LLH0 or LLH1 was applied before intensification. The consecutive perturbations with LLH2 of a ruin-recreate heuristic with any other choice between two mutation heuristics, LLH5 and LLH3, are very rewarding for solving instances of a bin packing problem as observed in Figure 6c,d. There is quite a balance between a single perturbation and consecutive perturbations for personnel scheduling, while solving the instances of the permutation flow-shop problem does not need extensive perturbations. The favored mutation heuristic of the EA-ILS hyper-heuristic in the domain is LLH0. Solving the problems of knapsack and maximum cut requires consecutive perturbations. Less than 25% of the best solutions found in the MAC6 and MAC7 instances are based on a single perturbation. This could explain why the EA-ILS hyper-heuristic was so effective on these problems unlike the FS-ILS and NR-FS-ILS hyper-heuristics. Although the effectiveness of the EA-ILS hyper-heuristic in BP, PS, KP, and MAC domains is evident, the LS-Seq procedure also contributed positively to its rapid navigation of the search space because it was able to quickly jump out of the local search procedure when a local optimum was detected.

## 5. Conclusions

The evolutionary algorithm-based iterated local search (EA-ILS) hyper-heuristic developed in this study is a highly effective tool for solving COPs. Hyper-heuristics are search methodologies for solving numerous forms of computational search problems with manifold applications in complex social problems affecting humanity. The EA-ILS hyper-heuristic recorded impressive performance when compared to the competitive hyper-heuristics for solving the QAP domain. It has obtained the best median ofv across several problem instances. The overall performance consolidates the observation that the EA-ILS hyper-heuristic obtained the best median ofv in 27 out of the available 30 instances across three supplementary problem domains. The reason that the EA-ILS hyper-heuristic generally obtained the best performance against FS-ILS and NR-FS-ILS is explained as follows: The three algorithms have the potential to pair perturbative LLHs when it is necessary. However, FS-ILS and NR-FS-ILS lack the feature of pairing two heuristics to visit a deeper search space because they are designed to operate based on the traditional perturbative LLH plus local search LLH obtained in a typical ILS-based implementation. The EA-ILS hyper-heuristic utilizes an evolutionary algorithm to automatically learn sequences of perturbative LLHs with the capability of pairing two heuristics for a deeper solution space search.

The EA-ILS hyper-heuristic has been presented in this paper as a high-performance tool for solving COPs with a focus on knapsack, quadratic assignment, and maximum-cut

problems. The evolutionary learning mechanism was employed to learn how an ILS solver perturbs the solutions while solving an instance of a given problem by an oscillating parameter of an existing acceptance mechanism. The EA-ILS hyper-heuristic has a different local search module that automatically learns the promising sequences of local search heuristics during the intensification phase of the algorithm. This study has made significant contributions to the existing discipline of combinatorial optimization and achieved the purpose of designing a hyper-heuristic for performance improvement. The extension of the existing ILS framework has been effective considering the dominating performance that the EA-ILS hyper-heuristic had over the existing hyper-heuristics according to the experimental results in QAP, KP, and MAC instances. The implementation of the EA-ILS hyper-heuristic has been able to correct the weaknesses of the ILS-based methods in the extended HyFlex domains by performing consecutive perturbations when necessary and utilizing a faster local search procedure, especially for problem domains such as personnel scheduling with slow heuristics. The EA-ILS hyper-heuristic did not completely dominate across the instances of HyFlex v1.0 but was able to outperform the FS-ILS hyper-heuristic across the six problem domains. In addition, it ranked ahead of the TS-ILS hyper-heuristic in the problem domains of PS and PFS in terms of effectiveness.

Reviewing the performance of the EA-ILS hyper-heuristic in the initial problem domains of HyFlex, it is evident that it struggled to quickly evolve effective perturbative heuristic sequences for some instances. Some domains where the EA-ILS hyper-heuristic struggled with evolving effective heuristic sequences include SAT, especially in SAT5 and SAT10, and TSP, especially in the TSP8 instance. This is because of the large amount time it takes the LLH to finish its operation on the TSP8 instance, while for the other instances, it could be the result of extreme criteria used for allowing an operation sequence into an archive. This backdrop opens up the need for more research efforts on the EA-ILS hyper-heuristic. A possible area of improvement could be to combine multiple criteria for archive entries, using the acceptance of solutions with the improvement of the best global solution instead of only the latter as employed in this study. Although the EA-ILS hyper-heuristic was competitive in both HyFlex 1.0 and HyFlex 2.0 problem domains, further improvement can be realized by combining an evolutionary algorithm to create heuristic sequences with effective selection mechanisms.

**Author Contributions:** Conceptualization, S.A.A. and O.O.O. (Oludayo O. Olugbara); methodology, S.A.A.; software, S.A.A.; writing—original draft preparation, S.A.A.; writing—review and editing, O.O.O. (Olufunke O. Oladipupo) and O.O.O. (Oludayo O. Olugbara); supervision, O.O.O. (Olufunke O. Oladipupo) and O.O.O. (Oludayo O. Olugbara). All authors have read and agreed to the published version of the manuscript.

**Funding:** This research received no external funding.

**Conflicts of Interest:** The authors declare no conflict of interest.

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
