# Peer review of "Evolutionary Algorithm-Based Iterated Local Search Hyper-Heuristic for Combinatorial Optimization Problems"

_algorithms, doi:10.3390/a15110405_

Round 1

Reviewer 1 Report

The present study offers a new selection hyper-heuristic, hybridizing an Evolutionary Algorithm (EA) and Iterated Local Search (ILS). The proposed approach, i.e. EA-ILS, is tested on 3 problem domains to show its effectiveness. 

- On critical missing element in the paper is that only 3 problem domains are considered while HyFlex originally has 6 more problems. This means that the empirical comparison should be made across 9 problem domains to validate whether the proposed hyper-heuristic is indeed a good one. Especially considering the generality aspect of hyper-heuristics and this study is based on HyFlex, it is natural to expect to see the results on those 9 problem domains.

- Referring to Section 4 (Experimental Results), it is mentioned that the performance values of the compared hyper-heuristics are taken from the corresponding websites / articles. I see that the runtime per instance is set as 530 seconds, considering the speed of their computer. By the way, please specify the computer setting used for the experiments. Additionally, it should be clarified how this 530 seconds runtime is set. Is it based on that program, provided by the CHeSC 2011 competition organizers, used to determine the 600 seconds comparable runtime? Even using that program won't be conclusive as it gives only approximate runtime values. This means that for any actual comparison, all the hyper-heuristics should be tested on the exact same computer configurations.

- I am a bit confused with Table 10. Is it testing the algorithm's parameters?  

- It would beneficial to release the code publically. 

- Section 5 (Conclusions) should be extended by the future research ideas.

- References might be extended, especially by the papers published in 2021 and 2022.   

Author Response

Reviewer 1

Comments to Authors

  1. The present study offers a new selection hyper-heuristic, hybridizing an Evolutionary Algorithm (EA) and Iterated Local Search (ILS). The proposed approach, i.e. EA-ILS, is tested on 3 problem domains to show its effectiveness. 

Response

We sincerely thank the reviewer for the positive comment about our work.

  1. On critical missing element in the paper is that only 3 problem domains are considered while HyFlex originally has 6 more problems. This means that the empirical comparison should be made across 9 problem domains to validate whether the proposed hyper-heuristic is indeed a good one. Especially considering the generality aspect of hyper-heuristics and this study is based on HyFlex, it is natural to expect to see the results on those 9 problem domains.

Response

We have followed the constructive comment of the reviewer and extended the application of EA-ILS to solve the 9 problems from HyFlex. Six from HyFlex v1.0 and the other originally solved problem domains are from the extended version.

  1. Referring to Section 4 (Experimental Results), it is mentioned that the performance values of the compared hyper-heuristics are taken from the corresponding websites / articles. I see that the runtime per instance is set as 530 seconds, considering the speed of their computer. By the way, please specify the computer setting used for the experiments. Additionally, it should be clarified how this 530 second runtime is set. Is it based on that program, provided by the CHeSC 2011 competition organizers, used to determine the 600 seconds comparable runtime? Even using that program won't be conclusive as it gives only approximate runtime values. This means that for any actual comparison, all the hyper-heuristics should be tested on the exact same computer configurations.

Response

The reviewer is right about the CHeSC 2011 organizers supplying a benchmark program to adequately gauge the capacity of a machine and hence the fair running time of a particular hyper-heuristic on a machine. The program (winbm.exe) was indeed run several times on the machine used for the experiments and it returns a time of 530 secs most of the time (therefore, the benchmark program was used to set the value of 530 secs), that’s why we decided to set the time limit as 530 secs. The specifications of the computer is an Intel core i5 system with an 8GB RAM. Yes, it could be faster, but we think some other essential programs running in the background and the flavour/generation/grade of the core i5 processor could be responsible for the higher run time.

Lately, we used a system with faster processing, using 484 secs as the time-based stopping criterion to test run EA-ILS on BP7 (an instance of the Bin Packing problem). We compared the results of the 31 solutions produced by the 530 secs run with that of the 484 secs run. We discovered that the performance was similar on the faster machine with 484 secs. We have provided the raw results obtained by both settings for 484 secs: https://github.com/dubystev/Synergy-HH/blob/master/BP7%20-%20Mstat2%20-%20484%20secs.txt

For 530 secs:

The faster the machine, the lower the time returned by the benchmark program, we want to give an assurance that the experimental tests on the algorithm was fair based on CHeSC 2011 rules.

  1. I am a bit confused with Table 10. Is it testing the algorithm's parameters?  

Response

This table is meant to show the less effort needed by the employed local search procedure (LS-Seq) and its comparative advantage over the VND-style local search invocation (LS-VND). Section 4.3 introduces what information we intend to convey in that table. The objective function values reported in the first two columns of the table indicate the performance comparison of using LS-Seq and LS-VND with the EA-ILS algorithm. The LS-Seq algorithm has been presented in section 3.2.4 – LS-Seq with illustration. Table 10 has changed to Table 14 because of the results of the 6 other problems added.

  1. It would be beneficial to release the code publically. 

Response

The source code has been hosted on the first author’s GitHub page at:

https://github.com/dubystev/Synergy-HH/blob/master/src/hh_project/EA_ILS_final.java

  1. Section 5 (Conclusions) should be extended by the future research ideas.

Response

This has been considered in the last two paragraphs of the conclusion section.

  1. References might be extended, especially by the papers published in 2021 and 2022.

Response

We have extended the paper's references with 6 recent ones (ref 13 to 18) published between 2021-2022. Changes in the first paragraph of the introductory section have reflected the more recent studies on hyper-heuristics.

Reviewer 2 Report

A hyperheuristic "EA-ILS" (evolutionary algorithm ILS) is proposed and applied to various COP benchmarks, showing good results. The paper is structured reasonably well, but I have concerns over the novelty and detail of the proposed approach.

Introduction: "This study was inspired by the inherent weaknesses exhibited by ILS hyper-heuristics..." despite the lengthy preceding list of related papers, the specific weaknesses are not clearly identified. This means it is difficult to see why the proposed new algorithm makes sense. The preceding list is largely descriptive with little critical analysis that could then be used to motivate the paper at this point.

"The EA-ILS hyper-heuristic was fortified with a local search module based on the connotation of the hidden Markov model" - it is not made clear where the HMM appears; as far as I can tell, it's a fairly simple EA.

Related studies: little to no attempt is made to connect previous work with the present study, or summarise key trends / gaps.

The ordering of the next few sections is a little odd; normally it makes more sense to present the new algorithm, then the experimental setup and results. Not necessarily a problem; but if the paper ends up being extensively reworked, it may be worth addressing.

Section 3.3.3: this appears to be the description of the major contribution: novel operators for the EA level of the HH. These need to be more clearly and formally described:

 - what is "wild mutation"?

 - add random: is this actually replacing the LLH at one or other position? 0 or 1 chosen uniformly at random? Or is it suggesting that a new LLH is added, and if there are already two, one is chosen UAR for replacement?

 - remove random: this seems to simply be uniform at random choice and removal of one LLH

The experimental results appear okay with suitable repeat runs and statistical analysis.

Author Response

Reviewer 2

Comments to Authors

  1. A hyperheuristic "EA-ILS" (evolutionary algorithm ILS) is proposed and applied to various COP benchmarks, showing good results. The paper is structured reasonably well, but I have concerns over the novelty and detail of the proposed approach.

Response

First, we sincerely thank the reviewer for the positive comment about our work. Second, we have provided more detail concerning the local search mechanism of the EA-ILS algorithm and extended its application to 6 other problems to show novelty. The novelty naturally comes in the unique application of evolutionary operators for effective perturbative heuristics to be discovered. Moreover, the experimental comparison of the EA-ILS hyper-heuristic with the existing hyper-heuristics on numerous COPs in the HyFlex framework has indicated the novelty of our algorithm. Finally, the source code has been hosted on the first author’s GitHub page at:

https://github.com/dubystev/Synergy-HH/blob/master/src/hh_project/EA_ILS_final.java

  1. Introduction: "This study was inspired by the inherent weaknesses exhibited by ILS hyper-heuristics..." despite the lengthy preceding list of related papers, the specific weaknesses are not clearly identified. This means it is difficult to see why the proposed new algorithm makes sense. The preceding list is largely descriptive with little critical analysis that could then be used to motivate the paper at this point.

Response

We have addressed this comment in the introductory section by discussing the weakness of ILS. The EA-ILS uses the LS-Seq procedure for the intensification stage. In terms of local search effectiveness, Table 14, section 4.4 compares the intensification/local search procedure of EA-ILS with a typical approach, the results in the table were meant to show that the proposed local search procedure is more effective across multiple domains. Even domains like PS, BP, and the initial three domains tackled showed why the approach is important to the success of EA-ILS. Section 4.5 further consolidates how the problem of lack of multiple perturbations in previous ILS-based approaches was addressed. We provided a thorough analysis of how EA-ILS perturbs solutions, revealing how multiple perturbations were important to its success in some domains.

  1. "The EA-ILS hyper-heuristic was fortified with a local search module based on the connotation of the hidden Markov model" - it is not made clear where the HMM appears; as far as I can tell, it's a fairly simple EA.

Response

The Local search module was not explicitly outlined in our paper, and we sincerely thank the reviewer for raising the issue. We have included its algorithm as outline in section 3.2.4. The local search procedure was inspired by previous work (a Sequence-based Selection hyper-heuristic, reference 70) which uses a hidden Markov model to model the relationships between the low-level heuristics. In this paper, it is applied to model the relationships between the local search heuristics because they have different neighborhoods. This enables effective sequences of local search heuristics to be automatically discovered over time and the discovered sequences are applied once the local search stage is invoked rather than random shuffling as done in the VNS approach.

  1. Related studies: little to no attempt is made to connect previous work with the present study, or summarise key trends / gaps.

Response

In the last paragraph of Section 2, we have attempted to address this comment in section 2 by highlighting the gaps based on the performances of previous hyper-heuristics on the domains investigated in this study.

  1. The ordering of the next few sections is a little odd; normally it makes more sense to present the new algorithm, then the experimental setup and results. Not necessarily a problem; but if the paper ends up being extensively reworked, it may be worth addressing.

Response

We have deleted the formulations of the problems in the paper.

  1. Section 3.2.3: this appears to be the description of the major contribution: novel operators for the EA level of the HH. These need to be more clearly and formally described: what is "wild mutation"?

Response

It is a term coined to imply that it is not a little change of a pair of heuristics (parent) that led to the formation of another pair of heuristics (child). The example cited for “wild mutation” has been incorporated into the paper.

  1. add random: is this actually replacing the LLH at one or other position? 0 or 1 chosen uniformly at random? Or is it suggesting that a new LLH is added, and if there are already two, one is chosen UAR for replacement?

Response

It means, adding a new heuristic to the sequence, which could lead to a replacement of an incumbent occupant of a randomly chosen position or a new addition entirely (if the second position is randomly picked, and it is not occupied).

  1. remove random: this seems to simply be uniform at random choice and removal of one LLH

Response

Removal of the randomly chosen position is the second one. Replacement of the element at the first position with the element at the second position if the position to remove from is the first position. But if the second position contains no element and the position to remove from is the first position, then a replacement is done.

  1. The experimental results appear okay with suitable repeat runs and statistical analysis.

Response

We sincerely appreciate the reviewer for the positive comment about the results.

Round 2

Reviewer 2 Report

I'm happy that my comments have been addressed.